# Conformational plasticity of ligand-bound and ternary GPCR complexes studied by $^{19}$F NMR of the β$_1$-adrenergic receptor

J. Niclas Frei[1], Richard W. Broadhurst [1], Mark J. Bostock [1,3], Andras Solt[1], Andrew J.Y. Jones[1], Florian Gabriel[1], Aditi Tandale[1], Binesh Shrestha[2] & Daniel Nietlispach [1]*

G-protein-coupled receptors (GPCRs) are allosteric signaling proteins that transmit an extracellular stimulus across the cell membrane. Using $^{19}$F NMR and site-specific labelling, we investigate the response of the cytoplasmic region of transmembrane helices 6 and 7 of the β$_1$-adrenergic receptor to agonist stimulation and coupling to a G$_s$-protein-mimetic nanobody. Agonist binding shows the receptor in equilibrium between two inactive states and a pre-active form, increasingly populated with higher ligand efficacy. Nanobody coupling leads to a fully active ternary receptor complex present in amounts correlating directly with agonist efficacy, consistent with partial agonism. While for different agonists the helix 6 environment in the active-state ternary complexes resides in a well-defined conformation, showing little conformational mobility, the environment of the highly conserved NPxxY motif on helix 7 remains dynamic adopting diverse, agonist-specific conformations, implying a further role of this region in receptor function. An inactive nanobody-coupled ternary receptor form is also observed.

[1] Department of Biochemistry, University of Cambridge, 80 Tennis Court Road, Cambridge CB2 1GA, UK. [2] Protein Sciences, CBT, Novartis Institutes for BioMedical Research (NIBR), Basel, Switzerland. [3] Present address: Biomolecular NMR and Center for Integrated Protein Science Munich at Department Chemie, Technical University of Munich, Lichtenbergstraße 4, 85747 Garching, Germany. *email: dn206@cam.ac.uk

G-protein-coupled receptors (GPCRs) are a family of plasma membrane-embedded, seven transmembrane sensors that respond to a wide range of extracellular ligands[1]. GPCRs act as signal transducers stimulating a response on the cytoplasmic side of the membrane via a range of intracellular binding partners (IBPs), including G proteins and β-arrestins[2,3]. With just over 800 GPCRs in humans they are the largest class of membrane proteins[4]. Signal transduction occurs following ligand binding, which leads to an allosteric conformational signal response, whereupon changes induced by the binding event at the extracellular side of the receptor drive conformational rearrangements at the cytoplasmic face, leading to coupling with IBPs[5]. Due to their central role in signal transduction GPCRs are involved in a wide range of physiological processes and consequently, are key proteins in many disease pathways[6]. Class A or rhodopsin-like GPCRs are the targets for around 1/3 of currently available drugs[6,7].

Receptor regulation of the signalling process occurs at multiple levels, involving binding of orthosteric agonists as well as allosteric modulators such as lipids[8]. Our understanding of GPCR signalling has been greatly enhanced by crystal structures of receptors in the inactive, ligand-bound and fully active (coupled to IBPs) states[9–11]. In addition to this static picture, NMR spectroscopy has demonstrated that these receptors are highly dynamic proteins, which populate multiple states that are in equilibrium with each other[12]. NMR has determined exchange kinetics and populations for some of these interconverting states[13,14]. Investigations of several class A receptors have revealed that their energy landscapes are unique with the differences relating to their individual signalling properties[15–21].

We investigate the conformational energy landscape of the β1-adrenergic receptor (β1AR). β-adrenergic receptors are central receptors in the sympathetic nervous system, which bind catecholamine ligands such as adrenaline or noradrenaline[22,23]. The β1AR is the predominant subtype in the heart, which is targeted by drugs such as β-blockers in the context of cardiac dysfunction[22].

Agonist-influenced equilibria between different receptor states of β1AR were recently described by $^{15}N$ and $^{13}C$ directed studies[16,17]. In our recent $^{13}C$ methionine NMR study, we presented evidence for equilibria between an inactive and pre-active state as well as for the existence of two interconverting ternary complexes[17]. Both equilibria were seen to be influenced by ligand efficacy, as evidenced by residues on transmembrane helices 5 and 6 (TM5 and TM6) located adjacent to the receptor binding pocket (M223[5.54], M296[6.41]) (superscripts refer to Ballesteros-Weinstein numbering[24]). These results corroborated and extended the identification of ligand-efficacy dependent equilibria and allosteric signalling in β1AR using $^{15}N$ valine-probes[16].

NMR studies to date have provided detailed descriptions of β1AR signalling using a range of probes. However, due to a lack of suitable reporters, only a limited description of the response of the important cytoplasmic region of the receptor to activation, crucial for IBP interaction, is available. Crystal structures of receptor ternary complexes of β1AR show the large amplitude outward movement of TM6, typical of activated GPCRs[11]. Previously, both $^{15}N$-labelled V280[6.25] [16] and $^{13}C$-labelled M283[6.28] [17] located at the cytoplasmic side of TM6 were observed to be insensitive towards ligand-based receptor activation, likely due to these probes facing the detergent micelle. Furthermore, there is little information about the role of the highly conserved NPxxY structural motif on TM7 for β1AR during receptor activation[25]. After a rotamer change following agonist activation, the side chain of Y[7.53] in the ternary complex forms a water-mediated hydrogen bond with Y[5.58] which stabilises the fully active state of the

receptor[26]. The TM7-helix 8 region has also been suggested to play a substantial role in the coupling specificity of both arrestins and G proteins[18,27].

We utilise the natural occurrence of C344[7.54] positioned after the NPxxY[7.53] motif at the end of TM7 in combination with fluorine-tagging to explore the cytoplasmic side of TM7 upon ligand binding and nanobody coupling (Fig. 1, Supplementary Fig. 1) using $^{19}F$ NMR. Furthermore, we introduce Cys282[6.27] at the equivalent position to C265[6.27] in the β2AR for a more detailed mechanistic investigation of the intracellular region of TM6 (Fig. 1, Supplementary Fig. 1). $^{19}F$ NMR studies have been widely used to investigate the conformational equilibria of GPCRs including the β2AR and A2AAR receptors[18–21,28–32]. Using these two reporter cysteines combined with an NMR active $^{19}F$-TET tag (Supplementary Fig. 2), we investigate the dynamic response of previously unexplored regions of the cytoplasmic face of the turkey β1AR to ligand binding and ternary complex formation, using a G protein-mimetic nanobody. Allosteric signal transduction from the orthosteric ligand-binding pocket on the extracellular side of β1AR to the cytoplasmic IBP interface, reveals a shifting equilibrium between inactive and active states, correlating with ligand efficacy. β1AR coupling to nanobody results in the formation of active as well as inactive ternary complexes, with the amount of active ternary complex correlating with the efficacy of the bound ligand. Hence, our study provides direct structural evidence for the formation of the active-state ternary complex in proportions reflecting the ligand efficacy, which in turn defines the subsequent level of downstream signalling β1AR. Interestingly, the conformational response in the two cytoplasmic regions on TM6 and TM7 of the ternary receptor complex is different. The response of TM6 indicates the formation of a single, well-ordered active-state ternary conformation in this region of the receptor, determined by the coupling partner, which is independent of the agonist type and shows little additional conformational dynamics. In contrast, TM7 in the vicinity of the NPxxY motif shows ligand-dependent conformational variability in the complex with extensive μs-to-ms dynamics and conformational features at the cytoplasm that are determined by the bound orthosteric ligand. Beyond the stabilisation of the active state, our observations suggest an involvement of the NPxxY microdomain during receptor activation in a manner that is predominantly related to the properties of the orthosteric ligand bound. This suggests a mechanistic role of this receptor region that might be independent of the coupled G protein, enabling the binding of further IBPs and resulting in a change in signalling bias or strength.

## Results

**Ligand binding conformational equilibria of TM6 and TM7.** To obtain information on the response of the cytoplasmic region of turkey β1AR to ligand binding, individual samples of β1AR-m-TM6-CysΔ2 (Fig. 2a, Supplementary Note 1, Supplementary Figs. 4, 6a) and β1AR-m-CysΔ2 (Fig. 2b, Supplementary Note 1, Supplementary Figs. 4, 5, 6b) solubilized in LMNG detergent micelles were investigated by 1D $^{19}F$ NMR upon addition of saturating amounts of agonists, including in order of increasing efficacy for TM6 atenolol, carvedilol, alprenolol, xamoterol, isoprenaline and for TM7 atenolol, carvedilol, alprenolol, cyanopindolol, bucindolol, xamoterol, isoprenaline and the natural ligand adrenaline (Supplementary Table 1). Bucindolol and carvedilol are known to be biased agonists for β1AR[33–36]. In the ligand-free apo state, A282C[TET, 6.27] on TM6 appeared as a single sharp peak, P1 (Fig. 2a). Only minor changes in chemical shift and linewidth were observed upon binding to the different agonists (Supplementary Fig. 6a, Supplementary Table 2). Although

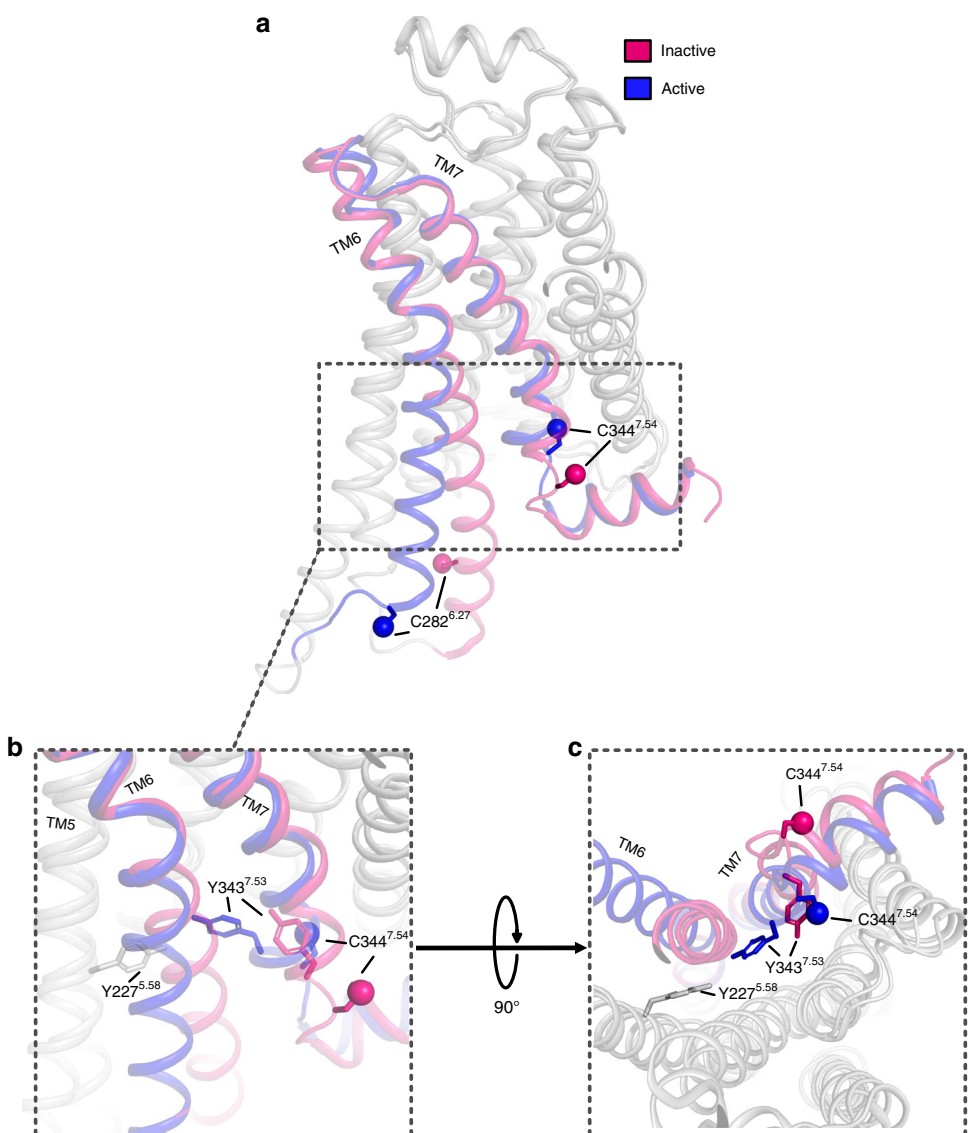

**Fig. 1 Structure overlay illustrating conformational changes in β₁AR upon activation.** **a** Side-on view of β₁AR bound to cyanopindolol, representative of the inactive state (magenta, PDB code 2YCY), and the nanobody Nb80 coupled receptor bound to isoprenaline, showing β₁AR in the fully active state (blue, PDB code 6H7J). TM6, TM7 and helix 8 are shown in the colour of their respective state, with the Gₛ mimetic nanobody omitted for purposes of clarity. The ¹⁹F tagged cysteines A282C$^{TET, 6.27}$ and $^{TET}$C344$^{7.54}$ are shown with their side chains represented as sticks and the Sγ atom as coloured spheres. The structure overlay highlights the outward movement of TM6 and the rotation of TM7 upon formation of the ternary complex. **b** Enlarged view showing the NPxxY$^{7.53}$ motif on TM7 containing the highly conserved Y343$^{7.53}$ that upon formation of the ternary complex rotates behind TM6 and makes a water-mediated hydrogen bond with Y227$^{5.58}$ on TM5 that stabilises the active state. Both Y343$^{7.53}$ and $^{TET}$C344$^{7.54}$ rotate inwards upon formation of the ternary complex. **c** Cytoplasmic view of the region shown in (**b**), which illustrates the clockwise rotation of Y343$^{7.53}$ and $^{TET}$C344$^{7.54}$ on TM7.

there was no systematic correlation with ligand efficacy, linewidths were slightly increased when bound to the higher efficacy agonists (Supplementary Fig. 7), suggesting some exchange broadening when bound to these latter ligands. For each of the current ¹⁹F spectra the signal P1 could be deconvoluted with a single Lorentzian line (Supplementary Fig. 8, Supplementary Table 2). Comparing the $R_2$ values of apo β₁AR and isoprenaline bound receptor substantiated the presence of a small exchange contribution to the linewidth of P1 (Supplementary Table 2). The failure to detect a more substantial response at the cytoplasmic side of TM6 parallels recent observations made in our ¹³C methionine NMR study using the immediately neighbouring residue on TM6, M283$^{6.28}$, as a probe[17]. Despite demonstrating via other residues e.g. M296$^{6.41}$ located on TM6 closer to the ligand-binding pocket that β₁AR-m responds in an efficacy

dependent manner to stimulation by agonists, the detergent exposed M283$^{6.28}$ showed only small changes[17].

To examine whether P1 was undergoing sub-second conformational exchange with other very low populated states, we conducted a series of saturation transfer experiments with the apo receptor and the isoprenaline bound receptor, respectively. We scanned a region from −1100 to +1100 (±2 ppm) relative to the main peak P1 in steps of 100 Hz but found no evidence of exchange.

Comparable ¹⁹F NMR experiments were conducted with β₁AR-m-CysΔ2 and agonists using $^{TET}$C344$^{7.54}$ on TM7 to probe the conformational response of the region immediately adjacent to the NPxxY$^{7.53}$ motif (Fig. 1). A single signal P2 (I₁ state) was observed for the receptor apo form that shifted gradually towards higher field as receptor samples were bound to

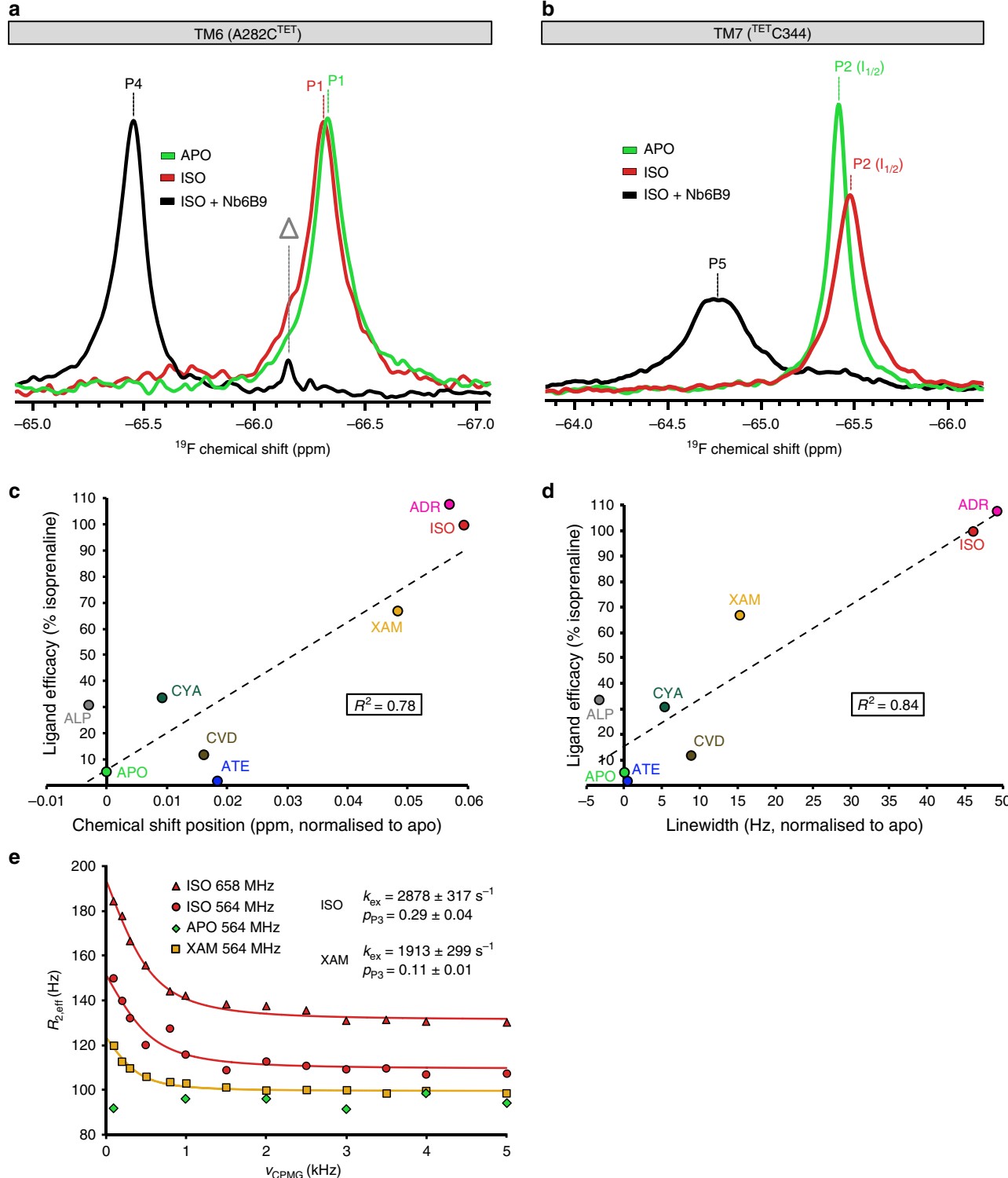

ligands of increasing efficacy (Fig. 2b, Supplementary Fig. 6b). Although the chemical shift changes were small (<0.06 ppm), they were accompanied by a considerable broadening of the signal linewidth on binding to agonists of higher $G_s$ efficacy (Fig. 2c, d, Supplementary Fig. 6b). This was particularly noticeable for xamoterol, isoprenaline and adrenaline, suggesting growing contributions to the receptor linewidth due to μs-to-ms timescale exchange. Signal positions and linewidths both correlated with the efficacy of the ligand bound (Fig. 2c, d). All signals for P2 were deconvoluted with a single Lorentzian line with no evidence

of multiple overlapping peaks (Supplementary Figs. 8, 20, Supplementary Table 2). Hence, the line broadening with the receptor bound to high efficacy agonists was further investigated through $^{19}$F NMR CPMG experiments that were performed at 564 and 658 MHz ($^{19}$F) and showed the effective removal of exchange broadening for $^{TET}$C344$^{7.54}$ as the pulsing rate $\nu_{CPMG}$ was increased from 100 to 5000 Hz (Fig. 2e). No measurable exchange contribution was detected for the apo receptor. By fitting the CPMG data for xamoterol and isoprenaline and reconstructing the corresponding spectral shifts of P2 relative to

**Fig. 2 $^{19}$F NMR spectroscopy of β$_1$AR TM6 and TM7.** The individual helices were studied by monitoring A282C$^{TET}$ (**a**) and $^{TET}$C344 (**b**), respectively, highlighting the response of the receptor to agonist binding and the formation of the ternary complex coupled to Nb6B9. All spectra were obtained at 308 K, 564 MHz ($^{19}$F) with receptor concentrations of 20 to 30 μM and saturating ligand concentrations (1 mM). **a** $^{19}$F NMR spectra are shown for A282C$^{TET}$ for apo β$_1$AR (light green), bound to full agonist isoprenaline (red) and with isoprenaline in ternary complex coupled to Nb6B9 (black). The peak corresponding to TM6 A282C$^{TET}$ in the apo and isoprenaline bound β$_1$AR (P1) appears at a chemical shift of −66.3 ppm. Addition of a two-fold molar excess of Nb6B9 causes a downfield shift of 0.8 ppm to −65.5 ppm (P4). The line marked with Δ indicates the appearance of free TET due to the slow cleavage of the S-S bond at 308 K. **b** $^{19}$F NMR spectra for TM7 C344$^{TET}$ show a peak at −65.4 ppm (P2) for the apo β$_1$AR (light green). Isoprenaline binding (red) causes an upfield shift by 0.06 ppm and a doubling of the linewidth. Coupling to Nb6B9 (black) shifts the signal of the ternary complex (P5) downfield to −64.8 ppm, together with a dramatic increase in the linewidth, compared to both isoprenaline bound and apo β$_1$AR. For $^{TET}$C344 on TM7 the chemical shifts (**c**) and the linewidths (**d**) of P2 correlate with the G$_s$ efficacy of the agonists tested. Chemical shifts and linewidths are shown relative to the apo receptor (δ = −54.42 ppm, Δ$ν_{1/2}$ = 46 Hz). Linear fits of the correlations are indicated by a line and R$^2$ values are given (ATE atenolol, CVD carvedilol, ALP alprenolol, CYA cyanopindolol, XAM xamoterol, ISO isoprenaline, ADR adrenaline). **e** Fast timescale μs-to-ms conformational dynamics of P2 assessed by $^{19}$F CPMG relaxation dispersion measurements for isoprenaline (red) (studied at $^{19}$F frequencies of 564 MHz and 658 MHz) and xamoterol bound receptor (orange) ($^{19}$F 564 MHz). Best fit curves to the dispersion data are shown together with values for $k_{ex}$ and $p_{P3}$ ($p_{I2}$) obtained from the fits. The apo form of β$_1$AR does not show any relaxation dispersion (green).

the apo receptor position, exchange parameters were determined for xamoterol ($k_{ex} = 1,913 \pm 299$ s$^{-1}$, $p_{I2} = 0.11 \pm 0.01$) and isoprenaline ($k_{ex} = 2,878 \pm 317$ s$^{-1}$, $p_{I2} = 0.29 \pm 0.04$) bound receptor (Fig. 2e, Supplementary Fig. 6b, Supplementary Table 2).

Applying the saturation transfer technique to a region surrounding the P2 peak (Fig. 3) a substantial drop in the P2 intensity revealed the presence of a previously invisible receptor signal, P3 (A state), at a position 300 Hz upfield from the corresponding signal P2 (Fig. 3c). With the saturating field positioned on P3, a saturation time course was recorded for the apo receptor and for the receptor bound to isoprenaline (Fig. 3d). A reference series with the position of the saturating field held at a distance 300 Hz downfield from P2 was also acquired. Both time intensity series for P2 were fit simultaneously and analysis gave an exchange rate $k_{ex} = 6.9 \pm 1.8$ s$^{-1}$ and a population $p_{P3} = 0.20 \pm 0.04$ for the isoprenaline bound receptor. For the apo form of the receptor the population of P3 dropped further to $p_{P3} = 0.15 \pm 0.04$, with $k_{ex} = 3.6 \pm 2.2$ s$^{-1}$. All fits of the saturation data required the transverse relaxation rate $R_2$ of the lower populated species P3 to be set to a substantially larger value ($R_2^{P3} = 600$ Hz) than for the main peak P2 ($R_2^{P2} = 80$ Hz for apo, $R_2^{P2} = 140$ Hz for isoprenaline bound). The broad linewidth explains why it had not been possible to observe the peak P3 directly by 1D $^{19}$F NMR (Fig. 3a, b).

**Formation of multiple ternary complexes with nanobody Nb6B9.** To observe the $^{19}$F NMR response of TM6 upon coupling to an IBP the apo receptor β$_1$AR-m-TM6-CysΔ2 was supplemented with a two-fold excess of Gα$_s$ mimetic nanobody Nb6B9. This revealed the appearance of a new, lower intensity signal P4 for A282C$^{TET, 6.27}$ at −65.4 ppm, shifted downfield by 0.9 ppm from the P1 signal of the ligand-free apo receptor, in addition to P1 (Fig. 4a). With the same two-fold excess of nanobody, the experiment was repeated with samples of receptor bound to one of the agonists carvedilol, cyanopindolol, xamoterol or isoprenaline, respectively (Fig. 4a). Each ternary complex showed a P4 signal of very similar chemical shift, with linewidths that were narrower than those of the corresponding P1 signal (Supplementary Table 2). All of the P4 signals could be deconvoluted as a single Lorentzian line (Supplementary Fig. 8, Supplementary Table 2). A steady rise of the P4 signal intensity with increasing efficacy of the receptor-bound agonist revealed the formation of more ternary complex (A$^{G+}$) with a concomitant decrease of the P1 signal intensity (Fig. 4a). The integrated relative signal area of P4 for the different nanobody complexes correlated well with the efficacy of the agonist bound (R$^2$ = 0.96) (Fig. 4c). Subsequently the same experiments were also repeated for carvedilol, xamoterol and isoprenaline with 5-fold and 10-fold excess of nanobody and

revealed that for these ligands, the ternary complex formation followed saturation behaviour, with the final amount of P4 determined by the agonist efficacy (Supplementary Fig. 9a).

The same series of measurements using a two-fold excess of nanobody was repeated with β$_1$AR-m-CysΔ2 to assess the response of TM7 $^{TET}$C344$^{7.54}$ upon Nb6B9 binding (Fig. 4b, Supplementary Fig. 21). For each of the agonists used the experiments showed a new peak P5 for $^{TET}$C344$^{7.54}$ that was shifted downfield from P2 by approximately 0.7 ppm and was attributed to the active ternary receptor complex (A$^{G+}$). However, in contrast to TM6, the TM7 region of the receptor in its apo form as well as when bound to the different agonists responded differently to nanobody binding, and resulted in P5 signals that varied in their position over a range of ~0.3 ppm, suggesting conformational differences on TM7 amongst the ternary complexes (A$^{G+}$) formed (Fig. 4b, Supplementary Fig. 9c, d). In contrast to the reduction in linewidth between P1 and P4 for A282C$^{TET, 6.27}$ (Fig. 4a, Supplementary Table 2), the signals of P5 for $^{TET}$C344$^{7.54}$ were substantially broadened when compared to their corresponding P2 signal (Fig. 4b, Supplementary Table 2). Again, the relative signal area of the active ternary peaks P5 showed excellent correlation with the efficacy of the ligands bound to the receptor (Fig. 4d), and a similar saturation behaviour as observed for TM6 upon addition of 5- and 10-fold excess of nanobody, respectively (Supplementary Fig. 9a).

Careful inspection of the signal area in the vicinity of the ligand-bound uncoupled receptor peak P2 near −65.5 ppm revealed the presence of an additional, broader peak P6 (A$^{G−}$) superimposed at a position very similar to P2 (Supplementary Fig. 9b). At the larger excess of nanobody the relative size of this broad P6 signal rapidly overtook the signal intensity of the sharper P2 signal of the uncoupled receptor. The presence of P6 was easily inferred from the increasingly broader appearance of the signal at the 5- and 10-fold excess of nanobody, indicating that the signal envelope at −65.5 ppm was increasingly dominated by the presence of more P6 as the remaining free receptor (P2) eventually bound to nanobody (Supplementary Fig. 9b). Except for isoprenaline the region around P2 required two Lorentzian signals for a satisfactory deconvolution resulting in a sharper component for P2 and a broader component for P6. The broader component became increasingly more intense in the presence of larger excess in nanobody (Supplementary Fig. 8, Supplementary Table 2). With the shift positions of P2 and P6 being very similar and due to a lower signal-to-noise ratio the deconvolution of the isoprenaline receptor region around P2 returned a single Lorentzian with a linewidth between the one of the sharper P2 and the broader P6 (Supplementary Table 2). Accordingly, all studies conducted in the presence of nanobody

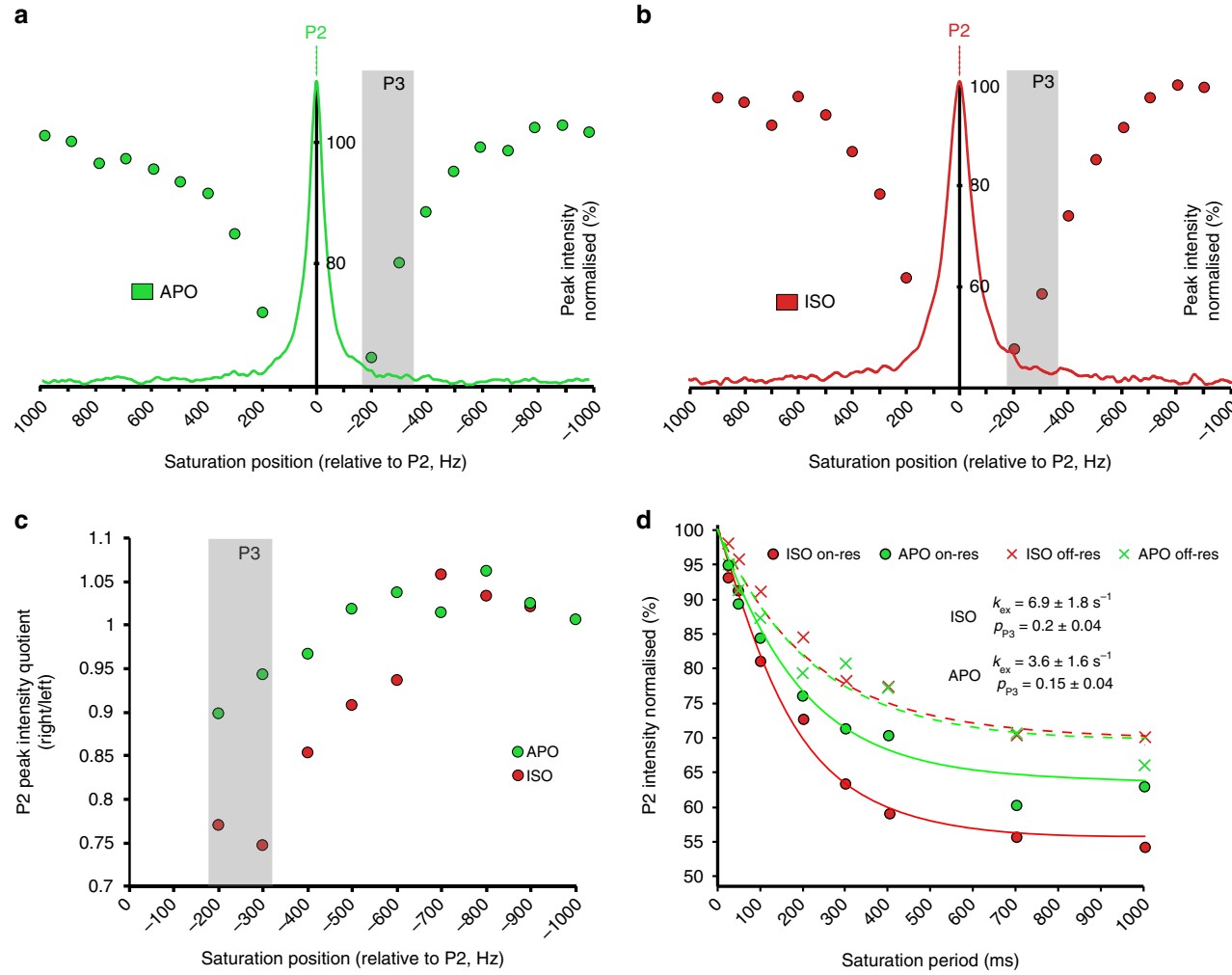

**Fig. 3 $^{19}$F saturation transfer experiments of $^{TET}$C344 on TM7.** The experiments identify a low populated signal P3 that is in slow exchange with P2. The offset dependence of saturation for (**a**) the apo receptor (green), and (**b**) $\beta_1$AR bound to isoprenaline (red) is probed in increments of 100 Hz. **c** Peak intensity ratios from pairwise experiments with saturation at symmetrical offsets reveal a maximal response at $-300$ Hz relative to the corresponding P2 signal (grey box). The response to saturation increases from the apo form to isoprenaline bound receptor. **d** Saturation time course for apo receptor (green) and $\beta_1$AR bound to isoprenaline (red) with the irradiation field (field strength 25 Hz) in the on-resonance experiment (circles) applied at the position of P3, i.e. $-300$ Hz from P2. For the off-resonance reference experiment (crosses) saturation was applied at $+300$ Hz from P2. Best fits for the signal decays are shown by lines with values for $k_{ex}$ and $p_{P3}$ extracted from simultaneous fits indicated for isoprenaline bound receptor as well as the apo form.

confirmed the presence of a second, alternative Nb6B9-coupled receptor form.

Saturation transfer was used to assess a possible slow sub-second exchange between the signals P5 and P6. A saturation transfer time course with the saturation field centred on P6 of TM7 $^{TET}$C344 (384 Hz upfield from P5) in the presence of a two-fold excess in Nb6B9 showed that P5 is in slow exchange with P6, with $k_{ex} = 8.1 \pm 1.6 \, \text{s}^{-1}$ (Supplementary Fig. 10a). A second pre-saturation experiment was conducted with the irradiation position further upfield from P6, at a sufficient distance (684 Hz upfield from P5) not to saturate P2 while still irradiating the broad foot of the P6 signal (Supplementary Fig. 10b). This resulted in a reduced but still noticeable response on P5 that is in agreement with the broad nature of P6 and confirmed the slow exchange process to be taking place between P5 and P6, with $k_{ex} = 8.0 \pm 1.5 \, \text{s}^{-1}$ showing a similar exchange rate (Supplementary Fig. 10).

Solvent-accessibility during ternary complex formation was assessed with a soluble $Gd^{3+}$ reagent (Supplementary Note 2).

## Discussion

We investigated the cytoplasmic region of $\beta_1$AR using $^{19}$F NMR via TET fluoro-tagging of the native C344[7.54] in TM7 and of a separately introduced A282C[6.27] in TM6. These reporters were used to study $\beta_1$AR in the apo form and with a range of agonists of varying efficacy (in order of increasing efficacy: atenolol, carvedilol, alprenolol, cyanopindolol, bucindolol, xamoterol, isoprenaline and the natural ligand adrenaline) as well as using a $G\alpha_s$ protein mimetic nanobody Nb6B9 to form ternary receptor complexes (Supplementary Table 1).

Our $^{19}$F NMR experiments show that the ligand-free $\beta_1$AR is mostly in an inactive state ($I_1$), as indicated by a single, well-defined signal, P1, for TM6 (Fig. 2a) and P2 for TM7 (Fig. 2b). This supports previous studies that focused on the transmembrane region of $\beta_1$AR near the ligand binding pocket and showed this region of the apo receptor to be much less dynamic than when bound to full agonist[16,17]. In view of the inherent low basal activity of the $\beta_1$AR, the apo form is likely to be representative of an inactive state, ($I_1$).

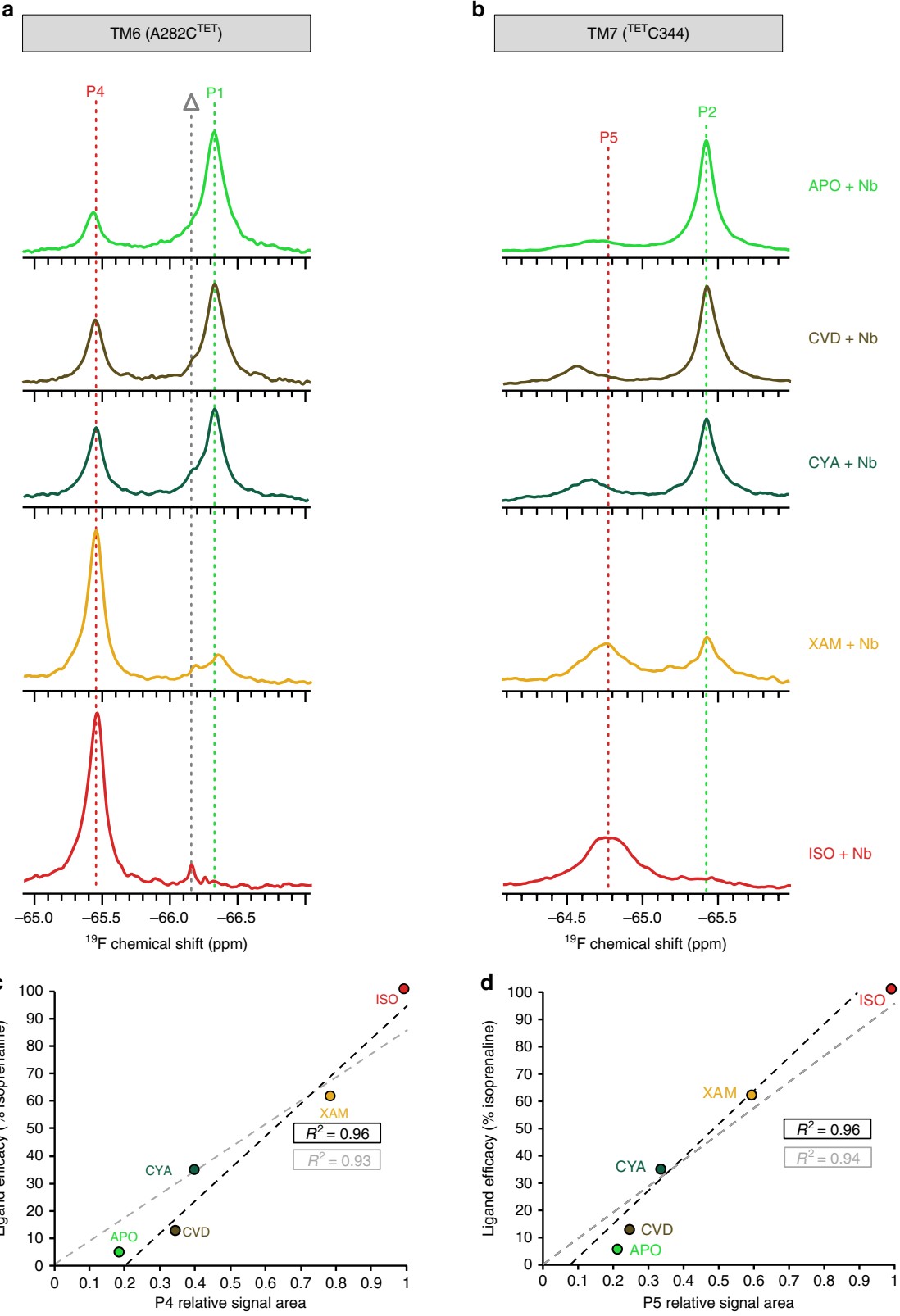

Probing the receptor with a range of agonists varying in efficacy revealed that the P1 signal for A282C$^{\text{TET, 6.27}}$ showed only a relatively small response to ligand binding (Supplementary Note 3, Fig. 2a, Supplementary Figs. 6a, 12, 13).

In contrast, while the TM7 signal P2 of $^{\text{TET}}$C344$^{7.54}$ indicated the apo receptor residing in an inactive form (I$_1$), the signal P2 gradually moved further upfield with increasing efficacy of the

bound agonist (Fig. 2b, c, Supplementary Fig. 6b). At the same time P2 also displayed increasing line-broadening (Fig. 2d). $^{19}$F CPMG relaxation dispersion experiments conducted at two fields for the receptor bound to isoprenaline revealed the presence of a fast exchange process with the receptor interconverting between two conformations, (I$_1$) and (I$_2$), with a rate $k_{\text{ex}} = 2,878 \pm 317\,\text{s}^{-1}$ (Fig. 2e). For xamoterol the exchange rate was reduced to

**Fig. 4 Ternary complex formation of β₁AR coupled to nanobody Nb6B9.** The experiments were conducted with the receptor in its apo form or in the presence of a range of agonists. All spectra were obtained at 308 K, 564 MHz ($^{19}$F) with receptor concentrations of $20 - 30$ μM, saturating concentrations (1 mM) of agonists and a two-fold molar excess of Nb6B9 over β₁AR. $^{19}$F NMR spectra of A282C$^{TET}$ (**a**) and $^{TET}$C344 (**b**) are shown for ternary complexes of β₁AR in the apo form (light green), bound to carvedilol (brown), cyanopindolol (dark green), xamoterol (orange) and isoprenaline (red) (CVD carvedilol, CYA cyanopindolol, XAM xamoterol, ISO isoprenaline). The spectra are shown from top to bottom in increasing efficacy of the ligand bound. For reference the position of the uncoupled apo form is shown by a green dotted line (P1, P2) and the position of the ternary nanobody coupled complex with isoprenaline as a red dotted line (P4, P5). In (**a**) the grey dotted line (Δ) indicates the appearance of TET due to slow cleavage of the S-S bond at 308 K. The relative integrals of the ternary complexes P4 (**c**) and P5 (**d**) linearly correlate with the ligand efficacy. With increasing efficacy the amounts of P1 and P2 are decreasing. In (**c**) and (**d**) a black dashed line indicates the linear fit to the measured data points, while a grey dashed line shows the corresponding linear fit that goes through the origin (based on the assumption that no ternary complex should be formed if the ligand efficacy is 0%).

---

$k_{ex} = 1,913 \pm 299\ s^{-1}$, while for the apo form of the receptor no additional exchange contribution to the linewidth was detected in the dispersion curve (Fig. 2e), in agreement with ligand-free β₁AR mostly populating an inactive state ($I_1$). However, for xamoterol and isoprenaline-bound receptor the second conformation ($I_2$) was increasingly populated. The low population of this second state ($I_2$) when in the apo form ($p_{I2} = 0.02$), with a slight increase when bound to xamoterol ($p_{I2} = 0.11$) and isoprenaline ($p_{I2} = 0.29$), indicates that ($I_2$) is likely a further inactive form of β₁AR (Fig. 2e). The timescale of the exchange process $I_1 \rightleftharpoons I_2$ is too rapid to relate to substantial conformational rearrangements of the TM7 environment as e.g. observed crystallographically in the transition from the inactive state (PDB code 2YCY) to the fully active state adopted in the ternary complex structure (PDB code 6H7J) (Fig. 1). Therefore, this process is most likely related to a smaller change in the side chain conformation, potentially a reorientation of the aromatic side chain of Y343$^{7.53}$, one of the conserved residues consistently observed to be involved in receptor activation and IBP coupling (Fig. 5)[11]. An increase in the population of ($I_2$) with strong agonists such as isoprenaline suggests that this conformational change might be linked to the early stages of receptor activation, with ($I_2$) possibly being an on-pathway intermediate to a further downstream activated state. Agonists would therefore shift the $I_1 \rightleftharpoons I_2$ equilibrium towards ($I_2$), facilitating subsequent activation.

In addition to the $I_1 \rightleftharpoons I_2$ equilibrium, the existence of a further state (A) in slow exchange with the inactive ($I_1, I_2$) states of the receptor (signal P2) was found indirectly through saturation transfer experiments, which revealed the presence of a strongly broadened peak at position P3, not directly detectable by $^{19}$F NMR (Fig. 3). While only investigated for the apo and the isoprenaline bound receptor it became clear that the population of this state increased with ligand efficacy from $p_{P3} = 0.15$ in the apo form to $p_{P3} = 0.20$ when isoprenaline bound. Substantial broadening of this relatively low populated state prevented its direct observation by 1D $^{19}$F NMR and suggested the occurrence of substantial amounts of μs-to-ms conformational dynamics in the vicinity of the NPxxY$^{7.53}$ motif once the receptor was bound to full agonists. We assume that the signal P3 corresponds to a pre-active receptor state (A) that is competent to bind nanobody, subsequently upon IBP coupling leading to the fully active state as seen in the structures of the ternary complex of the receptor (Fig. 5)[37]. The slow rate of exchange (ISO: $k_{ex} = 6.9 \pm 1.8\ s^{-1}$, APO: $k_{ex} = 3.6 \pm 1.6\ s^{-1}$) between the states ($I_1, I_2$) and (A) is indicative of major structural rearrangements taking place in this region of the receptor. The exchange process ($I_1, I_2$) $\rightleftharpoons$ (A) happens on a similarly slow sub-second timescale as observed in β₂AR for the move of TM6 away from the helix bundle when exchanging between inactive and active-like states, suggesting that cytoplasmic TM6 and TM7 rearrangements might be related and due to steric reasons occur in parallel[14]. Based on the reorientation of aromatic side chains in the immediate environment of

C344$^{7.54}$ we calculated an increase in shielding of 0.16 ppm due to changes in ring current shifts when moving from the inactive structure of β₁AR (approximated by the cyanopindolol bound structure) (PDB code 2YCY) to the ternary complex with Nb80 (as an approximation of the pre-active receptor state) (PDB code 6H7J), suggesting a move of $^{TET}$C344$^{7.54}$ into a more hydrophobic environment upon reaching the pre-active state (Supplementary Fig. 14, Supplementary Table 3). This is in reasonable agreement with our experimental observation of P3 at a position 0.5 ppm upfield from the apo receptor signal P2 (Fig. 3), and suggests that the slow exchange process relates to the rotation and inward movement of the intracellular part of TM7 that allows Y$^{7.53}$ following a change in the side chain rotamer conformation to slot behind the displaced TM6 (Fig. 1). As expected, the population of the (A) state increases with ligand efficacy, reaching 20% for isoprenaline. This is in a similar range to the 25% population found for the pre-active state using $^{13}$C methionine NMR, as assessed by the reporters M223$^{5.54}$ and M296$^{6.41}$[17]. At the same time the exchange rate seems to increase with ligand efficacy as well ($k_{ex}^{ISO} > k_{ex}^{APO}$) (Supplementary Fig. 2e).

Our observation of multiple inactive receptor states ($I_1, I_2$) for β₁AR agrees with several NMR studies and MD simulations on β₂AR[15,19,38–40]. The latter have suggested various intermediates between active and inactive states during deactivation simulations of β₂AR, which in some cases showed a TM7 conformation different from the inactive or active state[41]. Different metastable states for β₂AR were found, with TM6 adopting active-like outward as well as inward inactive-like conformations with TM7 not having reached its active-state like conformation.

MD simulations into the formation of a continuous internal water network during GPCR class A activation found such a network to be interrupted in the inactive state by the presence of two water-free layers of hydrophobic amino acid residues residing above the NPxxY$^{7.53}$ motif and below the conserved Y$^{7.53}$ that opened upon agonist activation to form a continuous water channel connecting the orthosteric binding site to the G protein interaction region[42]. Y$^{7.53}$ was found to transition between three rotamer conformations, representative of inactive (closed water channel), meta state (water channel closed at cytoplasm) and active state (open water channel). In the meta state, Y$^{7.53}$ remained in a hydrophobic layer that breaks upon reaching the fully active state as a continuous water channel is formed[42]. In agreement with a potential meta state and the postulated model of water accessibility, the environment of $^{TET}$C344$^{7.54}$ adjacent to Y$^{7.53}$ in our postulated pre-active (A) state is more hydrophobic than in the ($I_{1,2}$) states as suggested by the upfield shift of P3 compared to P2 (Fig. 3, Supplementary Fig. 14)[43]. This agrees with the suggestion that in the (A) state the hydrophobic layer next to Y$^{7.53}$ is still intact (Fig. 5). Accordingly, Y$^{7.53}$ in the (A) state might have already partly rotated into an alternative conformation while not yet reaching the fully active state. In agreement with our experimental data this process is bound to be slow

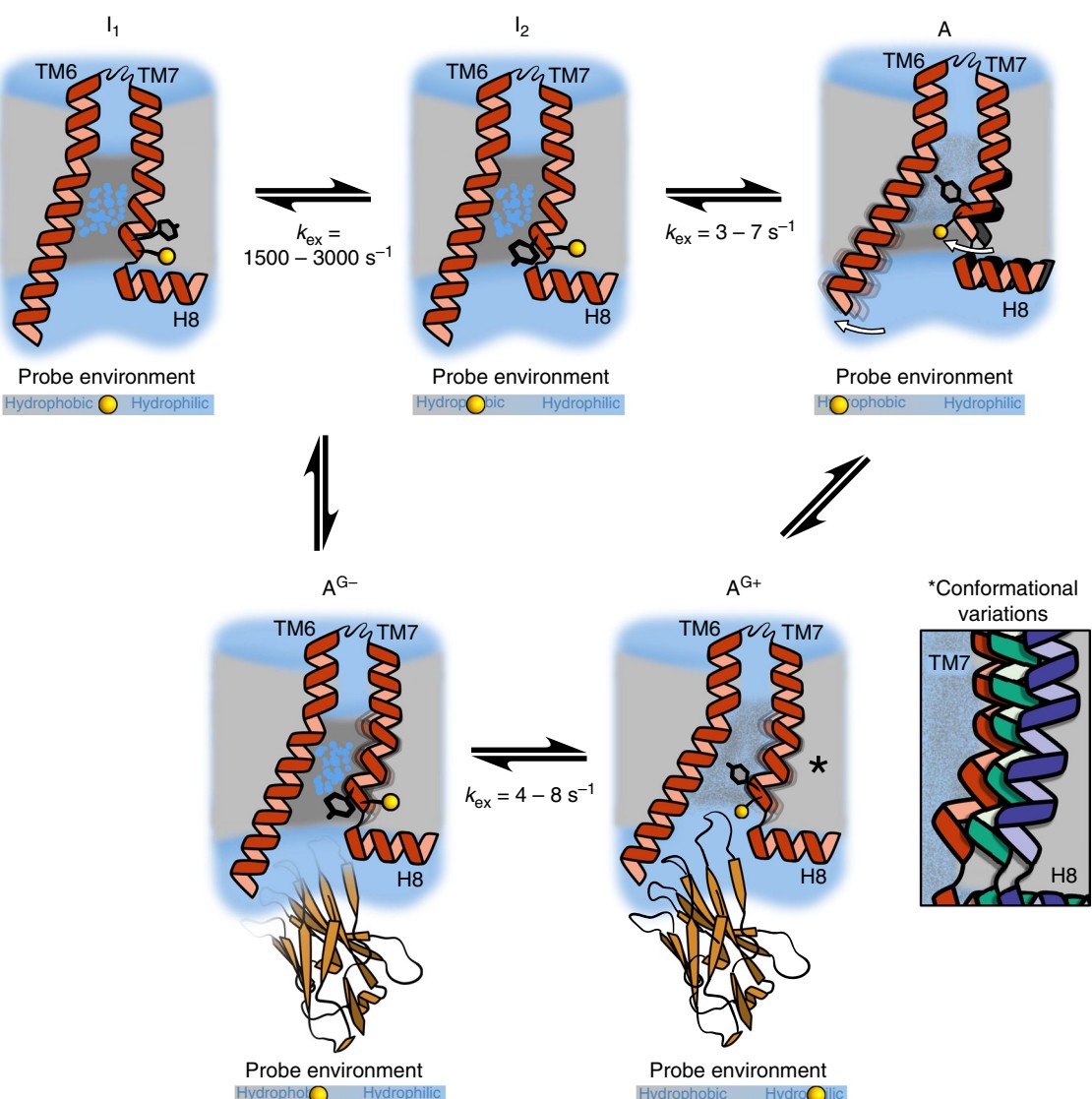

**Fig. 5 Schematic overview of β₁AR ligand activation and ternary complex formation.** Cartoon cross sections spanned by TM6, TM7 and H8 with the TM7 $^{TET}C344^{7.54}$ probes shown as yellow spheres, adjacent to $Y343^{7.53}$ of the NPxxY motif. The receptor exists in an equilibrium of inactive states ($I_{1,2}$) and a pre-active state (A), with the latter populated in growing amounts with increasing efficacy of the bound ligand. Nanobody Nb6B9 addition leads to the formation of a fully active ternary complex ($A^{G+}$) in amounts proportional to the ligand efficacy. Nanobody binding also occurs with the inactive form of the receptor, resulting in the formation of ($A^{G-}$). The latter can be considered as a pre-coupled inactive form, with inactive and active ternary complexes in slow exchange with each other. The binding interface in the ($A^{G-}$) complex is shown faded, emphasising that Nb6B9 has not yet fully engaged the epitope characteristic of the active receptor state. The ($I_1$) ⇌ ($I_2$) interchange takes place on a μs-to-ms timescale, while the ($I_2$) ⇌ (A) interchange as well as the ($A^{G-}$) ⇌ ($A^{G+}$) exchange occurs on a slower sub-second timescale. Exchange rates, where measured, are indicated below the equilibrium arrows. In the ternary state ($A^{G+}$) the cytoplasmic region of TM6 is rigid while TM7 remains dynamic on the μs-to-ms timescale, implied by TM7 showing partly blurred. In ($A^{G+}$) the conformation of TM7 in the vicinity of the NPxxY motif reveals agonist-dependent conformational differences, emphasised by TM7 showing in different colours in the enlarged region marked with (*). The grey slider below each receptor cartoon approximates the relative hydrophobic/hydrophilic extent of the TM7 $^{19}$F NMR probe surrounding in that particular state. Lipid bilayer hydrophobic regions are shown in light grey. Dark grey areas indicate transmembrane regions of the receptor with residues rich in hydrophobic side chains. These form hydrophobic gates above and below the NPxxY region (e.g. in $I_1$ and $I_2$) that shield the receptor interior against bulk water access. Blue dots on a grey background (e.g. in $I_1$, $I_2$ and $A^{G-}$) indicate ordered internal water molecules, separated from the bulk water through the hydrophobic gates (dark grey). Conformational changes upon activation disrupt the two hydrophobic side chain layers, resulting in the gradual opening of a continuous internal water pathway with cytoplasmic influx of bulk water, as indicated by the speckled grey/blue area between TM6 and TM7 in (A) and ($A^{G+}$).

as it involves the rotation and inward shift of TM7 and depends on the outward movement of TM6 having already occurred. Therefore, we suggest that the pre-active (A) state is competent to bind nanobody but has not transitioned into the fully opened conformation yet, retaining $Y^{7.53}$ (and accordingly $^{TET}C344^{7.54}$) in a hydrophobic environment.

Nanobodies such as Nb6B9 have been used to stabilise the active state of receptors[44,45]. In the β₂AR ternary complex they maintain the receptor in a conformation very similar to the one in the heterotrimeric $G_s$ protein-bound complex[46,47]. Due to the smaller size of the ternary nanobody complexes and their better stability in detergents, we used Nb6B9 for our $^{19}$F NMR studies of

$\beta_1$AR. Several crystal structures of $\beta_1$AR-nanobody complexes are now available (Supplementary Fig. 15)[37].

Addition of a two-fold excess of Nb6B9 to samples of $\beta_1$AR either bound to one of the diverse partial or full agonists used or in the receptor apo form led to the formation of an active state ternary complex ($A^{G+}$) for each of the samples investigated, as evidenced by the appearance of a new $^{19}$F NMR signal P4 for the TM6 probe (Fig. 4a) and a signal P5 for TM7 (Fig. 4b). The P4 and P5 peaks were shifted substantially downfield from the signals of the uncoupled receptors. We were unable to rationalise the observed large $^{19}$F chemical shift changes between the ternary complexes and the ligand bound receptors through ring current shift calculations that relied on the known structural coordinates (Supplementary Figs. 13, 14). It is likely, therefore, that the observed $^{19}$F shift changes result mainly from variations in the solvent exposure of A282C$^{TET}$ and $^{TET}$C344 when adopting the ternary state. Although chemical shift changes for $^{19}$F are difficult to predict, the effects of variations in polarity were investigated by Ye et al. where a downfield shift was found to be indicative of an increase in solvent polarity or augmented solvent exposure[43]. Full activation into the ternary state results in the outward movement of TM6 and rotation of TM7 with the inward movement of the NPxxY$^{7.53}$ motif (Fig. 1, Supplementary Fig. 16), that allows Y$^{7.53}$ to adopt the conformation that is unique to the active state by forming polar contacts with Y$^{5.58}$ on TM5 and water-mediated polar contacts to other residues, slotting behind TM6 (Fig. 1)[11]. The large downfield shifts observed upon formation of the ternary states suggest changes in the polarity of the local environment with A282C$^{TET}$ and $^{TET}$C344 experiencing greater solvent exposure in the fully active state of the ternary complex. This can be understood as A282C$^{6.27}$ moves outwards away from the TM core, while for C344$^{7.54}$, the hydrophobic layer adjacent to the cytoplasm is now opened forming a continuous water channel between the ligand-binding pocket and cytoplasm (Fig. 5). Further discussion on the solvent exposure of the $^{19}$F probe can be found in the supplementary information (Supplementary Note 4, Supplementary Figs. 11, 17, 18).

For each agonist-bound receptor sample, the relative amount of ternary nanobody complex ($A^{G+}$) formed was determined by integration of the corresponding NMR signal. For both A282C$^{TET}$ (signal P4) (Fig. 4c) and $^{TET}$C344 (signal P5) (Fig. 4d), the amounts of active ternary complex ($A^{G+}$) in solution correlated very well with the efficacy of the bound agonist in the complex. Although ligand efficacy values are typically derived from cell-based assays, our experiments illustrate that agonist $G_s$ directed signalling efficacy and the phenomenon of partial agonism closely relate to the total molecular amount of ternary receptor complex formed, as assessed here in vitro by two independent probes located at the cytoplasmic ends of TM6 and TM7.

For TM6, all active ternary complexes ($A^{G+}$) showed P4 at the same chemical shift position indicating a strong similarity in the conformational environment of A282C$^{TET}$ across the different complexes (Fig. 4a), regardless to which ligand the receptor was bound. The signals were sharp with narrower linewidths than for the ligand-only bound receptor signals P2 (Fig. 4a, Supplementary Table 2), indicating the absence of any dynamics near A282C$^{TET}$ that would broaden the NMR signal. This is in agreement with previous observations with full-agonist iso-prenaline bound $\beta_2$AR, which resulted in a sharper signal upon coupling to Nb80, as was assessed by a BTFMA probe attached to C265$^{6.27}$ [19]. Interestingly, our previous $^{13}$C NMR study on $\beta_1$AR indicated that the region of TM6 below the binding pocket was still mobile, with the dynamics showing a dependency on the type of ligand bound, as assessed by M296$^{6.41}$ [17] This suggests that even in the ternary state increased mobility persists in the regions of the allosteric network that are closer to the orthosteric binding

pocket. In contrast, based on our $^{19}$F NMR data, TM6 seems likely to be more ordered near the cytoplasm, once the conformation that couples with the nanobody has been engaged. Accordingly, a single conformation for TM6 near A282C$^{TET}$ is adopted (Fig. 4a). This agrees with the cytoplasmic side of TM6 in its active signalling conformation guiding the positioning of the IBP, while the type of binding partner dictates the extent of displacement of the cytoplasmic end of TM6 relative to the receptor core. Therefore, a more rigid arrangement at the cytoplasm is suggested to increase the efficiency of the signalling transfer onto the coupling partner, while the residual dynamics below the orthosteric binding pocket in the receptor core maintain the allosteric signal transmission initiated by the orthosteric ligand bound.

In contrast, the appearance of the signal P5 from the cytoplasmic region of TM7 in the same complexes varied over a wide range of 0.3 ppm with the position of P5 determined by the orthosteric ligand bound (Fig. 4b, Supplementary Fig. 9d). This indicates that $^{TET}$C344$^{7.54}$ reveals succinct ligand-dependent conformational differences in the adjacent NPxxY$^{7.53}$ motif and/or helix 8 amongst the different agonist-bound ternary receptor complexes. Furthermore, all the ternary complex P5 signals of TM7 $^{TET}$C344$^{7.54}$ were strongly broadened, supporting the presence of substantial µs-to-ms dynamics as this receptor region continues to sample multiple conformations (Fig. 4b, Supplementary Table 2). We tested whether the signal positions of P5 correlate with the efficacy of the ligands bound in the respective ternary complexes. A correlation would suggest that partial agonism is not only controlled via the amount of signalling complex formed but also via the adopted conformation at the cytoplasm of TM7. The weak correlation ($R^2 = 0.51$) between ligand $G_s$ efficacy and chemical shift position of P5, however, indicated that this is not the case (Supplementary Fig. 9c). Therefore, the observed conformational variability of the NPxxY motif on TM7 seems unrelated to $G_s$ protein signalling (Supplementary Note 5, Supplementary Figs. 15, 19).

The difference in the conformational response of TM6 and TM7 to Nb6B9 binding is remarkable and suggests a potential role for the TM7 NPxxY$^{7.53}$ motif that extends beyond the stabilisation of the fully active state. Possibly this might indicate a tendency for this receptor region to engage alternative signalling pathways, be related to biased signalling, be relevant for IBP coupling specificity or reveal an additional layer of $G_s$ independent signalling control that is determined by the orthosteric ligand. It is remarkable to note in this context that the carvedilol-bound complex resulted in the most downfield shifted peak and the largest difference to the balanced full agonist isoprenaline (Fig. 4b, Supplementary Fig. 9d). Previously, carvedilol has been associated with $\beta_1$AR biased signalling[33,35,36]. For the biased agonist bucindolol the effect is less pronounced with a shift similar to cyanopindolol. Accordingly, the biased agonist carvedilol might induce a conformation on TM7 of $\beta_1$AR upon $G_s$ protein binding that facilitates binding of further IBPs such as e.g. $\beta$-arrestin, leading to distinct cellular signalling outputs. Indeed, such complexes of GPCRs, $\beta$-arrestin and a G protein have been reported[48].

$^{19}$F NMR studies of the $\beta_2$AR have previously reported a semi-independent response of the TM6 and TM7 conformational equilibria following binding of orthosteric ligands of differing bias[18,49]. It was postulated that arrestin biased ligands preferentially activated TM7 over TM6, suggesting an involvement of TM7 in biased signalling. Our $^{19}$F NMR data for the ligand-bound $\beta_1$AR do not show such a response, possibly hindered through the inaccessibility of the P3 signal. In contrast, we observe a ligand dependent variability of the signal position for TM7 in ternary complexes of $\beta_1$AR (Fig. 4b) that seems largely decoupled from the

response of TM6. As for the $\beta_2AR$ this might indicate therefore that TM7 in $\beta_1AR$ plays a role in signal bias.

In order to assess the maximal amount of ternary complex that can be formed with nanobody we supplemented $\beta_1AR$ bound to carvedilol, xamoterol or isoprenaline, with a 2-fold, a 5-fold and a 10-fold excess of Nb6B9. Similar looking ligand-efficacy dependent saturation curves were obtained for TM6 and TM7 (Supplementary Fig. 9a). From our previous [13]C NMR study, it was known that at such an excess of nanobody there is no uncomplexed $\beta_1AR$ left in solution[17]. Hence, the P2 signal ($I_1$, $I_2$ state) of the uncoupled receptor should have vanished. Upon closer inspection of the spectrum near the position of the P2 signal of [TET]C344 we realised the gradual appearance of a new, broad peak P6 located in a very similar position to P2 that grew with the amount of Nb6B9 added (Supplementary Fig. 9b). In contrast the sharper, original P2 signal of the free receptor was gradually disappearing upon nanobody addition. Deconvolution of the P2/P6 region of the spectra using two Lorentzian signals confirmed the presence of the sharper P2 signal and the broader P6 signal (Supplementary Table 2), with the intensity of the latter increasing and P2 decreasing as more nanobody was added (Supplementary Fig. 8g, h). Due to its similarly broad lineshape to the active ternary signal, P5, we conclude that in P6 the receptor is also nanobody-bound and hence also in a ternary complex. However, its signal position, similar to P2, suggests an inactive receptor complex, possibly indicating pre-coupling of the receptor to nanobody. Very likely this signal is related to the minor active ternary form ($A^{G-}$) observed in our methionine [13]C NMR study[17]. This form should also be present for TM6, however, due to the lack of dispersion in this region and the similarity in linewidths between ternary (P4 signal) and ligand-only bound receptor (P1 signal), it is not as clearly manifested except for the xamoterol complex where it can be observed through an increase of the signal linewidth at the position of P1 compared to the ligand-only bound form (Supplementary Table 2). Using saturation transfer at two offsets we established a slow sub-second timescale exchange between P5 ($A^{G+}$) and P6 ($A^{G-}$) (Supplementary Fig. 10).

The results of our [19]F NMR study are summarised in a model shown in Fig. 5. Ligand binding acts as a functional modulator by shifting the populations between interconverting inactive ($I_1$, $I_2$) and pre-active (A) states, with the latter increasingly populated with higher efficacy ligands. Conformational exchange between inactive and pre-active states takes place on a sub-second timescale, while ($I_1$, $I_2$) exchange occurs on the $\mu$s-to-ms timescale. Pre-active but also inactive receptor states can couple to Nb6B9 forming two type of ternary complexes. The latter interconvert on a slow sub-second timescale. The total amount of active complex ($A^{G+}$) formed ultimately reflects the level of efficacy towards $G_s$ signalling for a given ligand. While the TM6 environment in the active ternary state is rigid, the TM7 environment remains dynamic. The active complexes with different ligands display conformational variability in the vicinity of the NPxxY[7.53] motif on TM7 (Fig. 5 inset), with possible functional implications. Based on concepts from MD simulations[42], we indicate two layers consisting predominantly of hydrophobic side chains that act as hydrophobic gates (Fig. 5, dark grey areas) surrounding the NPxxY motif. These gates are disrupted in the $A^{G+}$ state allowing the formation of a continuous water pathway from the ligand pocket to the cytoplasm. The hydrophobicity of the environment surrounding the TM7 probe varies between the different states, as indicated by the position of the yellow sphere on the slider below each structure cartoon (Fig. 5). $A^{G-}$ might represent a pre-coupled form of the receptor that can convert into the active state $A^{G+}$, its existence might be particular to the use of nanobody as a $G_S$ mimetic or be related to its binding affinity, in view of the

excess of nanobody over receptor used. In Fig. 5 the position of TM6 and TM7 in $A^{G-}$ is shown less open than in $A^{G+}$, to emphasise that the receptor has not adopted the fully active state yet.

In conclusion, we investigated the response of the under-explored cytoplasmic region of turkey $\beta_1AR$ to ligand activation and nanobody binding using [19]F NMR in combination with TET labelling of two cysteine probes located on TM6 A282C[6.27] and TM7 C344[7.54]. We show that allosteric signal transmission initiated by agonist binding leads to the population of different inactive and pre-active receptor states that are in equilibrium with each other, gradually shifting the receptor towards a more active state as higher efficacy ligands are used. Coupling of a $G_s$ protein mimetic nanobody to receptors bound to agonists of varying efficacy results in the formation of an active ternary receptor complex. The total amount of the latter relates to the efficacy of the ligand bound, indicating that signalling output is proportional to the amount of this species formed, providing a molecular link to the concept of partial agonism. The responses of TM6 and TM7 upon nanobody binding, however, are very different. The cytoplasmic side of TM6 is well ordered, determined by the IBP coupling partner, shows no evidence of $\mu$s-to-ms dynamics and reveals the same conformation for different agonists bound. In contrast, TM7 displays distinct conformational variability in the vicinity of the NPxxY motif that depends on the agonist bound and in addition is dynamic on a $\mu$s-to-ms timescale. These agonist-dependent conformational differences in the NPxxY motif in the ternary complexes suggest a further mechanistic role of this receptor region and indicate that the response of TM6 and TM7 are partly decoupled from each other even in the ternary states. The conformational differences at the cytoplasm of TM7 are particularly pronounced between the ternary complexes with unbiased agonists and the one with the biased agonist carvedilol, suggesting a functional role.

## Methods

**$\beta_1$AR construct generation.** The turkey $\beta_1AR$-m-Cys$\Delta$2 construct was modified from the previously published $\beta_1AR$-Met2-$\Delta$5 construct by introducing two cysteine mutations at position C85V[2.48] and C163L[4.47] [17]. Primer sequences are given in Supplementary Table 4. Differences between the $\beta_1AR$-m-Cys$\Delta$2 and the wildtype $\beta_1AR$ are truncations at the N-terminus, C-terminus and IL3, three thermostabilising mutations (R68S[1.59], E130W[3.41], F327A[7.37]), five methionine substitution mutations (M44L[1.35], M48L[1.57], M179L[EL2], M281A[IL3], M338A[7.48]) and four cysteine substitution mutations (C85V[2.48], C163L[4.47], C116L[3.27] for yield improvement and C358A to remove a palmitoylation site). The five methionine mutations are not required for this work but were maintained for comparison purposes with our earlier work[17]. The amino acid sequence for $\beta_1AR$-m-Cys$\Delta$2 is

MGAELLSQQWEAGLSLLLALVVLLIVAGNVLVIAAIGSTQRLQTLTNLFITS
LAVADLVMGLLVVPFGATLVVRGTWLWGSFLCELWTSLDVLCVTASIWTLC
VIAIDRYLAITSPFRYQSLMTRARAKVIILTVWAISALVSFLPIMLHWWRDEDP
QALKCYQDPGCCDFVTNRAYAIASSIISFYIPLLIMIFVYLRVYREAKEQIRKIDR
ASKRKTSRVAAMREHKALKTLGIIMGVFTLCWLPFFLVNIVNVFNRDLVPD
WLFVAFNWLGYANSAANPIIYCRSPDFRKAFKRLLAFPRKADRRLHHHHHH
HH.

The $\beta_1AR$-m-TM6-Cys$\Delta$2 construct was generated from the $\beta_1AR$-m-Cys$\Delta$2 construct through cysteine substitution C344S[7.54] and introduction of the non-native cysteine C282 through mutation A282C[6.27]. The amino acid sequence for $\beta_1AR$-m-TM6-Cys$\Delta$2 is

MGAELLSQQWEAGLSLLLALVVLLIVAGNVLVIAAIGSTQRLQTLTNLFITS
LAVADLVMGLLVVPFGATLVVRGTWLWGSFLCELWTSLDVLCVTASIWTLC
VIAIDRYLAITSPFRYQSLMTRARAKVIILTVWAISALVSFLPIMLHWWRDEDP
QALKCYQDPGCCDFVTNRAYAIASSIISFYIPLLIMIFVYLRVYREAKEQIRKIDR
ASKRKTSRVACMREHKALKTLGIIMGVFTLCWLPFFLVNIVNVFNRDLVPDW
LFVAFNWLGYANSAANPIIYSRSPDFRKAFKRLLAFPRKADRRLHHHHHHHH.

**$\beta_1$AR expression, purification and [19]F labelling.** All receptor expressions were performed using the FlashBac baculovirus insect cell expression system. Transfection reactions for baculovirus generation were prepared by mixing 0.2 $\mu$g of pBacPak8 plasmid containing the $\beta_1AR$ expression constructs with 4 $\mu$L Cellfectin II (Thermo Fisher Scientific), 2 $\mu$L FlashBac DNA (Oxford Expression Technologies) and 100 $\mu$L of Insect Xpress medium (Lonza). The transfection reaction was incubated for 30 min at room temperature and then applied directly onto adherent

Sf9 insect cells seeded at a density of $0.5 \times 10^6$ cells per mL in a culture volume of 1 mL. The transfection reaction was incubated at 27 °C for 5 days shaking at 90 rpm. After incubation, the cells were visually inspected for signs of viral infection. The resulting cell supernatant containing the recombinant baculovirus was harvested (P0 virus generation) and was used for further rounds of viral amplifications.

For expression, Sf9 insect cell cultures at a density of 1 to $1.5 \times 10^6$ cells per mL were infected with 4 mL virus stock per 1 L Sf9 cell culture. The expression was carried out for 48 to 72 h at 27 °C shaking at 160 rpm and the cells were subsequently harvested by centrifugation ($4000g$, 10 min, 4 °C). Resulting cell pellets were either used directly for $\beta_1$AR purification or stored at −20 °C.

Extraction of the $\beta_1$AR from insect cell membranes was performed by resuspension of the cell pellet with solubilisation buffer (20 mM Tris pH 8, 350 mM NaCl, 1% LMNG, 3 mM imidazole, Complete protease inhibitor (Roche)). The solubilisation was incubated for 1 h at 4 °C with stirring and the resulting suspension was cleared by centrifugation (175,000 $g$, 45 min, 4 °C). The cleared supernatant was applied onto a HisTrap FF 5 mL Nickel affinity column on an AKTA Pure (GE Healthcare) pre-washed with equilibration buffer (20 mM Tris pH 8, 350 mM NaCl, 0.02% LMNG, 3 mM imidazole). After protein binding, the column was washed in steps with the same buffer containing 3 mM, 50 mM and 250 mM imidazole, the latter eluting $\beta_1$AR.

The labelling of the $\beta_1$AR with the $^{19}$F probe 2,2,2-Trifluoroethanethiol (TET) was performed according to Supplementary Fig. 2[32]. In brief, the nickel affinity chromatography elution fractions were combined and concentrated (Amicon Ultra-15 spin concentrator with 50 kDa molecular weight cutoff) to 10 μM protein concentration. To activate cysteine side chains for $^{19}$F labelling, 4-4′-Dithiopyridine (commercially available as Aldrithiol-4™, Sigma-Aldrich) was added in a tenfold molar excess (100 μM) together with oxidised glutathione in a fivefold molar excess (50 μM). The solution was stirred at 4 °C for 20 min. After incubation, Aldrithiol-4™ and oxidised glutathione were removed by 1000× dilute concentration into 20 mM Tris pH 8, 350 mM NaCl and 0.02% LMNG. Following buffer exchange, 100 μM TET together with 50 μM oxidised glutathione were added and the solution was incubated for 30 min at 4 °C with stirring. The TET and oxidised glutathione were removed by 1000× dilute concentration into 20 mM Tris pH 8, 350 mM NaCl and 0.02% LMNG. The labelled $\beta_1$AR was further purified by alprenolol ligand affinity chromatography and the functional receptor eluted with either 1 mM atenolol or 0.1 mM alprenolol (Supplementary Figs. 21, 22).

Although not monitored routinely as untagged receptor is invisible to our investigations the efficiency of the $^{19}$F labelling reaction was estimated from small-scale receptor preparations to be on the order of 80–95%. The estimate is based on the intensity comparison of the $^{19}$F NMR signal of free TET, released from the receptor following reduction of the S–S bond with DTT, relative to an external TET standard and the intensity of the SEC A280 signal of $\beta_1$AR prior to the labelling reaction.

Similar labelling reactions were also attempted with the reagents 3-bromo-1,1,1-trifluoroacetone (BTFA) or 2-bromo-N-(4-(trifluoromethyl)phenyl)acetamide (BTFMA), respectively (Supplementary Methods, Supplementary Fig. 3)

**Expression and purification of Nb6B9**. The expression and purification of Nb6B9 followed established protocols[17]. In brief, the nanobody Nb6B9 was expressed in BL21-RIL *E.coli* cells and the cell pellets were lysed before clearing the lysate by centrifugation (75,600 $g$, 4 °C, 30 min). The cleared lysate was applied onto a HisTrap FF 5 mL Nickel affinity column on an AKTA Pure (GE Healthcare) pre-washed with equilibration buffer (20 mM Tris pH 8, 150 mM NaCl) and washed with the same buffer containing 15 mM imidazole before eluting the bound protein with 250 mM imidazole. The elution fractions were further purified by cation exchange chromatography using a RESOURCE S column (GE Healthcare) pre-washed with equilibration buffer (50 mM Sodium Acetate pH 4.8, 75 mM NaCl). The nanobody was eluted from the RESOURCE S column with a linear NaCl gradient ranging from 75 mM to 1 M NaCl. The pure Nb6B9 was finally buffer exchanged into 10 mM Tris pH 8, 150 mM NaCl and concentrated to approximately 1 mM protein concentration.

**NMR experiments**. NMR samples containing 5% $D_2O$ were prepared with ligand added to a final agonist concentration of 1 mM (atenolol, carvedilol, alprenolol, cyanopindolol, bucindolol, xamoterol, isoprenaline, adrenaline) to the apo form of $\beta_1$AR receptor (20–50 μM) solubilized in 1% LMNG. The population of the ligand-bound receptor exceeded 99.9%. Nanobody was added in two-fold molar excess for ternary complex formation, unless specified otherwise. NMR spectra were recorded at 308 K on a 600 MHz ($^1$H) and 700 MHz ($^1$H) Bruker Avance III spectrometer equipped with a 5 mm QCI HFCN/z cryoprobe ($^{19}$F 564 MHz) and a 5 mm TCI HCN/z cryoprobe ($^{19}$F 658 MHz), tuneable to $^{19}$F (The Francis Crick Institute, London). 1D $^{19}$F NMR data were obtained with a pulse acquire experiment recording 2560 complex points (50 ms), using a repetition time of 1 s and 5000 to 30,000 scans, resulting in a total experiment time varying between 2 and 12 h, ensuring that the signal-to-noise ratio was at least 30. To investigate the presence of μs-to-ms timescale dynamics leading to line broadening of P2, $^{19}$F CPMG relaxation dispersion measurements were recorded as a series of 1D experiments using a constant time implementation with a total transverse decay period of 10 ms, which allowed $\nu_{CPMG}$ to be increased from 100 to 5000 Hz ($\nu_{CPMG} = 1/(4^*\tau_{CPMG})$. Dispersion curves with 10 relaxation points were measured

at 564 MHz ($^{19}$F) and 658 MHz ($^{19}$F) (The Francis Crick Institute), which took 2 days per series. The consistencies of the samples were monitored over the course of the relaxation dispersion series by 1D $^1$H and $^{19}$F NMR, so that signal losses due to hydrolysis of the TET-tag or receptor degradation could be corrected for.

To investigate slow exchange on the chemical shift timescale between P2 and P3, and between P5 and P6, a series of 1D saturation transfer experiments was recorded at 564 MHz ($^{19}$F) where the initial excitation pulse was preceded by a 2 s recovery delay followed by a 1000 ms saturation period (saturation field strength of 25 Hz). For each of the experiments the position of the saturation field was incremented by 100 Hz, covering a range between −1100 Hz and 1100 Hz relative to the main signal of interest (P2 or P5). Comparing the pairwise intensities in the symmetrically irradiated experiments (on-resonance vs off-resonance (i.e. reference) experiment), the saturation offset with the strongest response was determined (for P3, saturation at −300 Hz relative to P2; for P6, saturation at −384 Hz relative to P5). Two time courses (8 points) were measured with the saturation length varying from 25 ms to 1000 ms and the saturating field positioned at ±300 Hz (P3), or ±384 Hz (P6) relative to the main peak. For P6 a second time course was measured with the saturation position changed to ± 684 Hz. Sample consistency was checked by 1D $^1$H and $^{19}$F NMR.

Solvent accessibility changes for $\beta_1$AR-m-Cys$\Delta$2 $^{TET}$C344 were assessed for the receptor in the presence of xamoterol (1 mM) and a two-fold excess in Nb6B9 through addition of increasing concentrations (0, 1, 3, 5 mM) of the $Gd^{3+}$ paramagnetic relaxation agent gadopentetic dimeglumine (Magnevist). Signal intensities and $R_2$ values for P2 and P6 were analysed as a function of $Gd^{3+}$ concentration. $R_2$ values were obtained from a two-point relaxation measurement by comparing the intensities in a CPMG experiment ($\nu_{CPMG} = 5000$ Hz) with the CPMG reference experiment.

All acquired FIDs were apodized with 20 Hz line broadening prior to zerofilling to 64k points and FFT using Topspin 3.1. $^{19}$F chemical shifts were calibrated with an internal standard of 2 μM trifluoroacetic acid (TFA) at −76.55 ppm relative to $CFCl_3$. All signals P1 to P6 were deconvoluted as Lorentzian lines to obtain the $R_2$ values related to the linewidth at half-height (Supplementary Fig. 8, Supplementary Table 2). $R_{2,eff}$ values from CPMG data were obtained from signal intensities according to $R_{2,eff} = (1/T) \times \ln[I_1(\nu_{CPMG})/I_0]$, with $T$ being the length of the constant time period (10 ms), $I_1$ the signal intensity with the 180° pulse train and $I_0$ the reference intensity in the absence of the constant time period. The errors in $R_{2,eff}$ were estimated from the noise in the spectra and from spectral variation. Fitting of the CPMG relaxation dispersion data and extraction of fitting parameters was done using in-house written software based on the methodology described by Baldwin[50]. The dispersion data was modelled as a two-state exchange process, with simultaneous fitting of the data recorded at two fields. Fitting was repeated for a range of offset differences $\Delta\omega$ between the exchanging states ($I_1$) and ($I_2$), ranking the results by their summed squared residuals. Combining the values from the best fit with the chemical shift changes of the P2 peak for different ligands, the exchange rate $k_{ex}$ for $I_1 \rightleftharpoons I_2$ and the populations of the exchanging states $p_{I1}$ and $p_{I2}$ were determined. Analysis of the slow-exchange saturation transfer experiments was done based on the Bloch–McConnell formalism using in-house written software to obtain $k_{ex}$ and $p_{P3}$ from simultaneously fitting the on- and off-resonance time courses[51,52].

**MD simulations**. The ternary structure model of $\beta_1$AR bound to xamoterol in complex with Nb6B9 (PDB ID 6H7N) was modified by coupling the cysteine side chain Sγ positions of C344 and A282C to TET and prepared for MD simulations with the Schrödinger 6 protein preparation wizard under the OPLS_2005 force field. The structure was embedded in a fully hydrated POPC bilayer coupled to TIP3P water molecules, using the OPLS3 force field for building the system and the steepest descent algorithm for energy minimisation. MD simulation time was set to 1 ns at constant volume and temperature, followed by 1 ns at constant pressure and temperature for initial system equilibration, while further extending the simulation time to 12 ns at 300 K.

**Reporting summary**. Further information on research design is available in the Nature Research Reporting Summary linked to this article.

## Data availability
The authors declare that relevant data supporting the findings of this study are available within the article and its Supplementary Information files or on reasonable request from the corresponding author. The source data underlying Figs. 2–4 and Supplementary Figs. 10, 11 are provided in a Source Data file.

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

## Acknowledgements

This work was funded through a Hershel Smith studentship to J.N.F. and a BBSRC research grant to D.N. (BB/K01983 X/1). F.G. is the recipient of a DAAD scholarship. A.T. is grateful for financial support from the Trinity Henry Barlow Foundation, Newnham College and the Cambridge Trust. We are grateful for access to the Biomolecular NMR facility of the Department of Biochemistry (U. Cambridge) and the MRC Biomedical NMR Centre of The Francis Crick Institute (London).

## Author contributions

J.N.F. and D.N. designed the research. J.N.F. made constructs, performed molecular biology, expressed and purified proteins and prepared N.M.R. samples. A.S., B.S., A.J.Y.J., F.G. and A.T. expressed and purified proteins. M.J.B. conducted chemical shift calculations. J.N.F. and D.N. collected and processed N.M.R. data and analysed the spectra. R.W.B. wrote in-house software for data analysis and performed relaxation data fitting. D.N., J.N.F. and M.J.B. prepared the manuscript. D.N. supervised the project.

## Competing interests

The authors declare no competing interests.
