## [Peer Review File · Nature Communications]

Reviewers' comments:

Reviewer #1 (Remarks to the Author):

Frei et al. present an interesting, meticulously prepared and discussed study on turkey β 1-adrenergic receptor using ^{19}F NMR to probe dynamics and population interconversion of the intracellular region using two probes: one at the N-terminal of TM6 and one just C-terminal to the NPXXY motif and proximal to helix 8. One might argue that two sites may not be sufficient to provide a great deal of insight; however, I think the results are rather exciting and strongly serve to expand knowledge of GPCR activation mechanisms while also bringing to light some very interesting hypotheses worth testing and expanding upon in future studies. Also very interesting is the demonstration that partial agonism may be directly reflected in the quantitative amount of ternary complex directly determinable through integration of the appropriate NMR signal. This may prove very valuable in and of itself. As a whole, I think this is an exciting and well supported study. Many of the constructs and conditions employed have been previously validated, allowing for a focus on ^{19}F NMR studies and their interpretation. Some minor comments which I would suggest be incorporated are detailed below.

1) All studies are carried out in micellar systems. It would be beneficial to provide additional rationale and justification for this choice of model membrane environment, over and above amenability to solution-state NMR studies.

2) With the lengthy focus in the results section on the labelling scheme used and others that did not work, some discussion would be merited (e.g., potential reasons that BTFA and BTFMA were less selective than the TET). An estimate of the efficiency of labelling achieved would also be beneficial, either via quantitative NMR methods or, perhaps, through high-resolution MS. Methods for the BTFA and BTFMA protocols employed are also not detailed – at the very least, it would be beneficial to detail this in the Supporting Information.

3) In Figs. 2 and 4 a hydrolysis product is noted. This should be explicitly noted and discussed in the text, not simply a note in the figure caption.

4) p. 7, para. 1, line 3 – typo “correlated both with”

5) Fig. 2e – the fit for XAM does not look spectacular. This could be repeated and/or carried out at a second field strength as with ISO in order to improve these data. As it stands, the error of the fit is rather large and it is challenging to state with certainty that there is a difference between XAM and ISO with respect to the relaxation dispersion profiles.

6) p. 9, para 1, line 10 – typo “at the position similar to”

7) Fig. 5 – this is a very nice summary figure! One additional expansion that may be of further benefit would be inclusion of timescales for the various interconversion steps.

8) Is there an estimate of T_1 for the TET probe in the GPCR-conjugated state? (A 1 s d_1 seems relatively rapid, but maybe this is perfectly sufficient for this particular probe?)

Jan K. Rainey, Dalhousie University, Halifax, Canada

Reviewer #2 (Remarks to the Author):

Frei et al. present a ^{19}F -NMR study of the turkey β 1-adrenergic receptor in complex with agonists and in tertiary complexes with agonists and a G protein mimicking nanobody. The NMR spectra are of high quality, exhibiting strong signal-to-noise, which is arguably better than the signal-to-noise

of data from some other ^{19}F NMR studies in the literature, and the presented data show striking chemical shift differences between binary and tertiary complexes. The series of spectra of tertiary complexes that exhibit changes in intensities correlating with ligand efficacy are quite nice and, to my knowledge, a novel observation that adds important information to the body of literature on this topic. Thus I support the publication of this work, provided that revisions are made to the manuscript, the following questions are addressed, and data from a couple control experiments are included in a revised version.

The results of helix VI are intriguing. While at most very minor differences are observed between the apo receptor and binary complexes with agonists, a new signal with a large chemical shift difference of ~ 1 ppm is observed upon addition of nanobody to the sample with an intensity that correlates with the efficacies of the various studied agonists. As noted by the authors, the lack of aromatic amino acids proximal to the helix VI ^{19}F probe could potentially explain the observation of little to no response of the NMR signal to the addition of agonists to the sample. However, a lot of effort appears to be directed at convincing the reader why ring current shifts do not explain the large chemical shift differences for the new signals observed with tertiary complexes. It would improve the paper if emphasis was shifted toward providing a rationale for this observation rather than what arguments do not explain the current observations.

Regarding the large chemical shift difference for the new signal observed with tertiary complexes for the individual probes in helices VI and VII, the authors suggest in the Discussion section that the large change in chemical shift between the binary and tertiary complexes may be due to differences between the hydrophobicity of the environment surrounding the NMR probe. In support of this argument, the authors reference a study from 2015 from the Prosser group that used model small molecules and calculations to claim that changes in the hydrophobicity of local environments around ^{19}F probes can account for large observed changes in chemical shifts. However, these calculations and experimental data from the 2015 paper do not provide sufficient explanation in the present study, especially because the largest sensitivities to changes in hydrophobic environments from the 2015 Prosser paper were for ^{19}F NMR probes containing an aromatic group, which is not present in TET. Support of the above argument would be greatly advanced by experimental data that tested this hypothesis, for example, by including NMR data of the tertiary complex where a soluble paramagnetic reagent is added to the sample.

Further, an alternative explanation that should be explored more is the possibility that the chemical shift difference observed for the tertiary complex is due to the proximity of side chains of the nanobody to the ^{19}F NMR probes. Upon examination of a published structure of $\beta 1\text{AR}$ in a complex with a nanobody and agonist (PDB 6H7N), this reviewer noticed that Ala 282 of $\beta 1\text{AR}$ is only 3-4 Å away from the nearby residue Ser 27 of the nanobody. Also given that TET would further extend the side chain of the modified Cys at position 282, it seems reasonable to suggest that direct interactions between the nanobody and TET-labeled Cys 282 could account for the observed chemical shift changes of the new signal for tertiary complexes. Indeed, Phe 29 of the nanobody is also then potentially close enough that ring current effects from the nanobody may play some role. The same argument applies also to the ^{19}F NMR probe in Helix VII, which also appears to be nearby several side chains of the nanobody. Also, a figure of the tertiary complex of $\beta 1\text{AR}$ with agonist and nanobody showing the location of the NMR probes should be included in the text.

How did the authors assign the NMR signals? Specifically, how did the authors rule out that C124, C133, and C302 did not contribute to the observed NMR signals? While these are buried in the hydrophobic region, labeling of such buried cysteines has been observed in the literature before (see, for example, Susac et al. *Angew. Chem.* 2015). As a control, the authors should show ^{19}F data where C282 is replaced with a different amino acid and the "background" signals shown. Also, ^{19}F data should be shown with C344 replaced by the endogenous Ser.

How was the functionality of the proteins used in the current study determined? The authors state

"The functionality of the construct was shown previously", referencing previous work (Solt et al 2017), but in the Solt paper there does not appear to be new functional data reported. Rather, in the 2017 paper reference is made to the work by Isogai et al. that tested the functionality of a similar, but not identical, protein. Further, there does not appear to be any functional data presented in the current manuscript that demonstrate the modified constructs used for the current study bind ligand and are functional. Some data should be included to show that the proteins used for ¹⁹F NMR are functional (e.g. radioligand binding experiments of the proteins employed in the current study).

The noted observation that the conformational equilibria of the helix VI probe and helix VII probe are at least semi-independent is interesting and analogous to similar observations made for receptor-ligand complexes in earlier ¹⁹F NMR studies of β 2AR (e.g. Liu et al Science 2012, Eddy et al Structure 2016). The text describing the current observations (middle of page 16 in the Discussion) would benefit from discussing the present observations in the context of this literature data.

Additionally, the following minor points and questions should be addressed in a revised version of the manuscript.

The title should be revised to more accurately reflect the study. Specifically, the use of "GPCR signaling complexes" should be revised, as these aren't actually signaling complexes studied but rather receptor complexes with small proteins that mimic some of the interactions observed with receptor-G protein complexes. I suggest the authors remove "signaling" from the title. This distinction is important especially as other groups (and likely the present authors) are also studying such receptor G protein complexes.

Supplementary Fig 2 – please include a reference to the PNAS Khorana paper where this scheme is first reported.

Supplementary Fig 3 – the samples of β 1AR actually appear to be quite heterogeneous. This reviewer is somewhat surprised that such samples provided clean NMR data, though it does not call into question the quality of the presented data. Some comments from the authors on this in the text would be appreciated.

For the samples where no ligand is added (apo) and a nanobody is added: how does the signal intensity of the tertiary complex compare with the known reported basal activity of the protein?

All the presented NMR spectra were recorded at 308K. Were data at lower temperatures measured and, if so, is there any evidence of multiple components in the signal envelopes?

Supplementary Figure 8 – what is the x axis shown here? This should be labeled.

In several spectra there is a reported "degradation product" (e.g. spectrum of Iso + Nb6B9 in Figure 2 panel a). What is this specifically?

Figure 4 – my opinion is that this would be more graphically pleasing to visualize if the order of the presented spectra were reversed.

Page 18 Discussion, "The cytoplasmic side of TM6 is well ordered, determined by the IBP coupling partner, shows few dynamics and reveals . . ." "shows few dynamics" is imprecise and not proper language for this manuscript. Please revise or remove completely.

Reviewer #3 (Remarks to the Author):

The paper reports data on the dynamic structural equilibrium of the beta1-adrenergic receptor and addresses the underlying energy landscape in response to agonist and nanobody binding. While the structural basis of GPCR-ligand and G-protein binding is becoming more complete due to the advances in crystallography, in particular in cryo-EM, the dynamic aspects that lead to the activation of GPCRs represent the key to understand the biological function of these highly important molecules on a basic level. The current study addresses exactly this requirement using NMR, which is the method of choice to investigate the dynamics of biomolecules. The authors have done a tremendous amount of work to characterize the conformational equilibrium of the beta1-adrenergic receptor in the apo form as well as in the presence of agonists with varying efficacy as well as in the presence of a nanobody. They used ¹⁹F labeling of two Cys residues in TM 6 and 7 to specifically study the dynamic structural alterations in the intracellular side of the receptor by this highly sensitive NMR method. Furthermore, they fully exploit their label using an impressive set of ¹⁹F NMR methods to come up with a comprehensive picture of beta1-adrenergic receptor activation. Overall, I am very impressed with this manuscript and think it is very well suited for publication in Nat. Commun. I have a few thoughts the authors may want to consider when they prepare a revised version of this highly interesting and important article.

The authors should comment why they (had to) use(d) a 33 to 50 fold excess of ligand over receptor to produce the (very small) chemical shift differences in the ¹⁹F NMR spectra.

The effective removal of the exchange broadening for C344 in the CPMG experiments is very intriguing. Did the authors also confirm that no relaxation dispersion is measured for site A282C? I am not so sure about the occurrence of the Peak P6 of C344 in the presence of the nanobody. If I am correct, this is the position of the apo or agonist bound state of the receptor. As the nanobody is added, this peak shifts significantly, why would there be two conformations for the apo/agonist bound receptor in addition to the new conformation that is observed in the presence of the nanobody? Regardless, it would be good to show the deconvolution/simulation of these spectra with 2 peaks. Nevertheless, it is hard for me to comprehend that in P6, the receptor is also nanobody-bound as the downfield shift of the signal is completely missing. This conclusion should be reconsidered.

In principle I like the schematic hydrophobic/hydrophilic bar on Fig. 5. But I would not agree that in state A the probe sees a maximally hydrophobic and in state AG+ a maximally hydrophilic environment. In general, I do not understand why the authors do not picture the interior of the receptor as hydrophobic (gray), why is there a hydrophilic environment in the middle?

Hydrophobicity and hydrophilicity are bulk properties, water channels and water mediated hydrogen bonds in my opinion cannot be considered as "phase".

Point-by-point reply to the reviewers' comments

Reviewer #1

We would like to express our gratitude to the reviewer Jan Rainey for his very favourable response to our manuscript and his enthusiastic endorsement of our work. We value the comments fully aimed at improving the quality of our manuscript and in the following we respond to the reviewer's comments.

1) All studies are carried out in micellar systems. It would be beneficial to provide additional rationale and justification for this choice of model membrane environment, over and above amenability to solution-state NMR studies.

We thank the reviewer for his comment regarding the membrane mimetic used in our study. The reviewer raises an important point here as one would like to see future studies conducted in as native-like membrane mimetics as possible. These should really be lipid containing. However at the moment, the vast majority of GPCR NMR studies in the literature have been conducted in detergent micelles (for a recent review see for example Bostock et al. *Curr. Opin. Struct. Biol.* 57, 145-156 (2019)). While the ideal environment would be as close to native as possible this currently remains very challenging in terms of sample preparation and spectroscopy. Attempts are underway to study these proteins in lipid-like membrane mimics but this is still not a routine procedure suitable for the sorts of measurements presented in this study using quantitative approaches such as CPMG or saturation transfer techniques. Our laboratory has started to conduct similar investigations with receptors embedded in more representative native-like lipid bilayer environments such as saposin nanoparticles and MSP nanodiscs. In terms of NMR spectroscopy such work is more demanding than using LMNG as the overall size of the solubilized receptor is increased. However, in the future with better samples in hand such work will decidedly be within the scope of what can be achieved with ¹⁹F solution NMR spectroscopy.

For reasons of comparability to our previous study using ¹³C NMR (Solt et al. *Nat. Commun.* 8, 1-12 (2017)) we decided to conduct all our investigations in this manuscript in the same LMNG detergent used by us previously. Our intention was to make our two investigations as similar as possible and to limit changes introduced to differences in the receptor constructs. In the first paragraph of the results section on pg 5 we have now added a sentence to clarify our choice of membrane mimetic. As such our two studies are highly complementary and the results can be compared with highest confidence. Changing only one key parameter at the time (i.e. here a slight change of the receptor constructs used) when comparing two studies seems to be also within the remits of good scientific practice.

2) With the lengthy focus in the results section on the labelling scheme used and others that did not work, some discussion would be merited (e.g., potential reasons that BTFA and BTFMA were less selective than the TET). An estimate of the efficiency of labelling achieved would also be beneficial, either via quantitative NMR methods or, perhaps, through high-resolution MS. Methods for the BTFA and BTFMA protocols employed are also not detailed – at the very least, it would be beneficial to detail this in the Supporting Information.

We thank the reviewer for his comment. In contrast to the reactions with BTFA or BTFMA, TET labeling is conducted via a two-step process and possibly it is the initial activation step using 4-4'-Dithiopyridine (4-DPS), which makes the overall ¹⁹F labeling reaction more selective. Possibly 4-4'-Dithiopyridine is less accessible to any of the other 'internal' cysteine sites. However, this is speculative, and we have not conducted any exhaustive investigation to further substantiate this suggestion. Ultimately, we are not sure why BTFA and BTFMA are less selective than TET and we certainly would not want to propose an extension of the validity of our current observations onto other GPCRs. Obviously, other groups have shown the

suitability of these reagents for the labeling of other receptors. Hence, in the absence of a sound explanation we would rather hold back from including further speculative comments on this topic in the manuscript.

We have now included the labeling conditions for the BTFA and BTFMA reactions in the Supplementary Information as a Supplementary Note on pg 26. The added procedures build on the protocols described by the Prosser group for BTFA (Chung et al. *J. Biol. Chem.* 36305-36311 (2012)) and BTFMA (Ye et al. *Nature* 533, 265-268 (2016)).

Although the ^{19}F labeling efficiency was not routinely measured we conducted three small-scale tests with $\beta_1\text{AR}$ and estimated the labeling efficiency to be on the order of 80-95%, with no noticeable difference between TM6 and TM7 labeling. The estimate is based on the intensity comparison of the ^{19}F NMR signal of free TET, released from the receptor following reduction of the S-S bond with DTT, relative to an external TET standard and the intensity of the SEC A280 signal of $\beta_1\text{AR}$ prior to the labeling reaction. This is now included on pg 21 in the Methods section where we describe the ^{19}F labeling. We added also that this test of the labeling efficiency was not done on a routine basis, as any unlabeled receptor is invisible in our studies.

3) In Figs. 2 and 4 a hydrolysis product is noted. This should be explicitly noted and discussed in the text, not simply a note in the figure caption.

We thank the reviewer for pointing this out. The observed product is simply the appearance of the TET signal over time due to slow cleavage of the S-S disulphide bond at 308 K. We acknowledge that the use of the terminology ‘hydrolysis product’ in this context is misleading and technically also wrong and have corrected this (see paragraph below). As some of the experiments are run over several days we see the release of free TET, due to the limited stability of the S-S bond. The cleavage reaction is slow compared to our experiment times and hence this is not a problem in our work, particularly as the affected receptor is simply becoming invisible to our investigations leaving just the occurrence of a sharp signal for TET. Where experiment series are concerned, we interleave the time series with reference experiments, which allows an estimate of how much breakdown took place over time. We did this during the CPMG and saturation transfer experiments as described in the methods section. Over the experimental time frame investigated, the rate of breakdown is linear, so that during quantitative series this slow breakdown can easily be corrected for.

In the legends to Fig. 2, Fig. 4, Supplementary Fig. 6 and Supplementary Fig. 8 we have now removed any reference to ‘hydrolysis’ and describe this now correctly as S-S cleavage that leads to the release of TET. In addition, we now also point out in the main text on pg 6 that slow breakdown of the S-S bond is taking place due to the elevated temperature at which the measurements are conducted and that sometimes this leads to the release of small amounts of TET. We indicate also that this does not interfere with any of our investigations.

4) p. 7, para. 1, line 3 – typo “correlated both with”

We have changed this on pg 7 to “both correlated with”.

5) Fig. 2e – the fit for XAM does not look spectacular. This could be repeated and/or carried out at a second field strength as with ISO in order to improve these data. As it stands, the error of the fit is rather large and it is challenging to state with certainty that there is a difference between XAM and ISO with respect to the relaxation dispersion profiles.

We appreciate the reviewer’s comment on the limited quality of the xamoterol CPMG data. We have now

repeated this CPMG measurement at 14.1 T (^{19}F 564.6 MHz) using a more concentrated receptor sample (80 μM in $\beta_1\text{AR}$). As can be seen from the updated Fig. 2e the quality of the relaxation dispersion data is now much better, resulting in a significantly improved fit and accordingly reduced errors. The quality of the xamoterol data is now at least as good as the isoprenaline data. Overall the additional measurement has corroborated our initial observation as such that there is a difference in exchange kinetics between xamoterol and isoprenaline bound receptor. Due to the better data we can now convincingly propose that the exchange rate $k_{\text{ex}}^{\text{XAM}} < k_{\text{ex}}^{\text{ISO}}$ and that the population of I_2 $p_{I_2}^{\text{XAM}} < p_{I_2}^{\text{ISO}}$. We have updated Fig. 2e with the results of the new measurement and have updated the k_{ex} and p_{I_2} values on the figure and in the text on pg 7 and pg 12. As the new measurement confirms our initial observations our interpretation remains unchanged. Therefore we have not made any further adjustments in the text regarding these observations.

6) p. 9, para 1, line 10 – typo “at the position similar to”

We have changed this on pg 9 to “at a position similar to”

7) Fig. 5 – this is a very nice summary figure! One additional expansion that may be of further benefit would be inclusion of timescales for the various interconversion steps.

We thank the reviewer for this suggestion. We have now included the timescales into our revised Fig. 5.

8) Is there an estimate of T_1 for the TET probe in the GPCR-conjugated state? (A 1 s d1 seems relatively rapid, but maybe this is perfectly sufficient for this particular probe?)

The $\beta_1\text{AR}$ T_1 values vary in the range 0.25 to 0.3 s both for ligand bound as well as the ternary ligand bound and nanobody coupled receptors. The differences in T_1 are small and are hardly affected by the additional coupling to the nanobody as the relatively small Nb6B9 only leads to a very minor increase in the overall rotational correlation time of the detergent solubilized receptor. Accordingly, the recovery delay of 1 s corresponds to $3.3 - 4 \times T_1$. A waiting period of $5 \times T_1$ would correspond to a ~fully relaxed measurement. While our chosen conditions are not far off the latter value, these settings are a compromise intended at increasing the signal-to-noise ratio per unit time. We would like to emphasize that this does not affect the interpretation of our 1D experiments nor the CPMG measurements. The most critical experiments in terms of obtaining reliable signal intensities are the saturation transfer experiments and these were conducted with a 2 s recovery delay. This is already stated in the NMR experiments section in the Methods on pg 23.

Reviewer #2

We thank the reviewer for his/her favorable comments on our manuscript and for making the many very valuable suggestions, recommendations and raising several important points.

1) The results of helix VI are intriguing. While at most very minor differences are observed between the apo receptor and binary complexes with agonists, a new signal with a large chemical shift difference of ~ 1 ppm is observed upon addition of nanobody to the sample with an intensity that correlates with the efficacies of the various studied agonists. As noted by the authors, the lack of aromatic amino acids proximal to the helix VI ^{19}F probe could potentially explain the observation of little to no response of the NMR signal to the addition of agonists to the sample. However, a lot of effort appears to be directed at convincing the reader why ring current shifts do not explain the large chemical shift differences for the new signals observed with

tertiary complexes. It would improve the paper if emphasis was shifted toward providing a rationale for this observation rather than what arguments do not explain the current observations.

We thank the reviewer for this comment. While ^{19}F NMR spectroscopy offers dramatic insights into conformational interchange of biomolecules such as GPCRs, as undertaken in our study, it is also widely appreciated in the field that understanding the factors that influence ^{19}F chemical shifts is still in its infancy compared to other nuclei. This is particularly the case when it comes to a more quantitative interpretation of chemical shift changes, for example when attempting to relate chemical shift differences to molecular rearrangements. This has been reviewed to some extent in Kitevski-LeBlanc et al. (Prog. Nucl. Magn. Reson. Spectrosc. 62, 1-33 (2012)) and only limited progress has been made on the topic since the appearance of the review. Chemical shift changes thus are mostly discussed qualitatively, however, effects related to ring current shifts, solvent accessibility or changes in hydrophobic environment on ^{19}F are amongst the factors that are becoming increasingly better understood.

In our study, a noticeable downfield change in chemical shift is observed between the binary and ternary receptor complexes, both for TM6 and TM7, and accordingly in our manuscript we were attempting to evaluate the possible causes.

In view of recent studies from the Wuthrich group that showed relatively convincingly the impact of ring current shifts on ^{19}F tagged GPCRs (Susac et al. PNAS 115, 12733-12738 (2018); Liu et al. J. Biomol. NMR 65, 1-5 (2016)) we assessed the possibility of such a similar scenario occurring for $\beta_1\text{AR}$ using available crystallography coordinates of inactive and active receptor states. The assessment for TM6 and TM7 based on the static structure information is briefly described in the manuscript on pg 15 and is supported by Supplementary Fig. 12 (TM6: comparing $\beta_1\text{AR}$ with $\beta_2\text{AR}$ environments), Supplementary Fig. 13 (TM6: comparing experimental values and calculated predictions for $\beta_1\text{AR}$), Supplementary Fig. 14 (TM7: comparing experimental values and calculated predictions for $\beta_1\text{AR}$), Supplementary Fig. 16 (TM7: comparing $\beta_1\text{AR}$ with $\beta_2\text{AR}$ environments) and Supplementary Table 3. The conclusion of our careful analysis is that the aromatic ring current shifts are not the decisive factor for the substantial downfield shift changes observed upon formation of the ternary complexes of $\beta_1\text{AR}$. Despite the large number of supplementary figures required for the support of our conclusion, in the manuscript text this analysis is kept very concise.

While the ring current analysis is not providing an explanation for the observed shifts in our study, we believe nevertheless that it is worthwhile to point out this noticeable difference in view of the previous reports on $\text{A}_{2\text{a}}\text{AR}$ and $\beta_2\text{AR}$. In the manuscript we then move swiftly onto considering other factors that might contribute towards the observed shift changes. In contrary to the reviewer's statement we do already spend considerable emphasis on considering other factors in view of finding a rationale that explains our chemical shift observations.

We discuss these effects in detail on pg 15 where we relate the observed chemical shift changes to alterations in the hydrophobic environment experienced by the ^{19}F probes associating these with changes in the water pathway that connects the orthosteric ligand binding pocket with the cytoplasm during receptor activation. Based on this rationale we do already provide an alternative comprehensive explanation for the observed shift changes. Following the reviewer's suggestion (reviewer point 2) we have now also conducted additional PRE measurements using the Gd^{3+} Magnevist reagent. The results of these additional investigations highlight the increased solvent-accessibility of the ternary complex compared to the ligand-bound receptor (for a detailed discussion see this reviewer point 2) and support our initially proposed suggestion of a change in polarity of the probe environment as a major contributor to the observed ^{19}F shift changes.

2) Regarding the large chemical shift difference for the new signal observed with tertiary complexes for the individual probes in helices VI and VII, the authors suggest in the Discussion section that the large change in

chemical shift between the binary and tertiary complexes may be due to differences between the hydrophobicity of the environment surrounding the NMR probe. In support of this argument, the authors reference a study from 2015 from the Prosser group that used model small molecules and calculations to claim that changes in the hydrophobicity of local environments around ^{19}F probes can account for large observed changes in chemical shifts. However, these calculations and experimental data from the 2015 paper do not provide sufficient explanation in the present study, especially because the largest sensitivities to changes in hydrophobic environments from the 2015 Prosser paper were for ^{19}F NMR probes containing an aromatic group, which is not present in TET. Support of the above argument would be greatly advanced by experimental data that tested this hypothesis, for example, by including NMR data of the tertiary complex where a soluble paramagnetic reagent is added to the sample.

We thank the reviewer for this comment. The stated study conducted by the Prosser lab (Ye et al. *J. Biomol. NMR* 62, 97-103 (2015)) showed that changes in polarity can result in large variations in ^{19}F chemical shifts. The authors demonstrated this based on experimental data with some of the typically used small molecular ^{19}F NMR probes as well as theoretical calculations. As the reviewer already says, the aromatic moiety containing tags such as BTFMA showed the largest effect. However, as shown for example in Fig. 2 of that study the TET probe used by us (which in the 2015 study is named TFET) displayed an effect which over the hydrophobic range investigated was at least 75% of that for BTFMA. This suggests that in principle substantial chemical shift variations due to changes in hydrophobicity can also be expected for non-aromatic group containing probes such as TET, depending on the polarity 'window' under consideration. Manglik et al. in their study of $\beta_2\text{AR}$ (Manglik et al. *Cell* 161, 1101-1111 (2015)) used BTFMA as a fluoroprobe attached to C265 (equivalent position to A282C in $\beta_1\text{AR}$) and observed ^{19}F shift changes $\Delta\delta$ of 1.3 ppm between the inactive state and the ternary state of $\beta_2\text{AR}$ coupled to Nb80. This is a larger shift change than observed in our study using TET, which at a qualitative level at least seems to be in agreement with the increased sensitivity of BTFMA to changes in polarity, as shown by Ye et al. 2015. Accordingly, our observed shift changes seem to be in the range of possible values based on changes in polarity of the ^{19}F probe environment.

We have now performed a series of solvent-accessibility experiments that used a water-soluble relaxation reagent as suggested by the reviewer. The study was conducted using the paramagnetic reagent Magnevist and with the $\beta_1\text{AR}$ TM7-labelled construct we performed a titration where we increased the concentration of the Gd^{3+} relaxation reagent. We added increasing amounts of Magnevist (0 – 5 mM) to $\beta_1\text{AR}$ in the presence of 1 mM xamoterol (to ensure fully bound receptor) and a two-fold excess of nanobody Nb6B9. Under these conditions (and in the absence of the Gd^{3+} reagent) the NMR spectrum looked as shown in Fig. 4b, indicating the presence of xamoterol-bound receptor (P2, 40%) and nanobody-coupled ternary receptor (P5, 60%). The concurrent presence of both receptor states allowed the side-by-side assessment of the impact of the relaxation reagent on the P5 and P2 signals, which increased the robustness of the experimental protocol. We assessed the effect of adding the paramagnetic relaxation reagent by measuring the signal intensities of P5 and P2 and their R_2 rates from a two-point measurement where we compared the signal intensities in a CPMG experiment ($\nu_{\text{CPMG}} = 5,000$ Hz, constant time of 2.5 ms) with a reference experiment. Observed changes in intensities, as well as R_2 rates, revealed a distinctly stronger response for the P5 signal to the addition of Magnevist (Supplementary Fig. 11). We interpret this result as an increased solvent-accessibility of ^{19}F C344 on TM7 in the ternary state compared to the ligand-bound state.

Accordingly, the relaxation enhancement study further strengthens the suggestion that the observed chemical shift changes between the ligand bound and ternary complex receptors are related to differences in the hydrophobic environment at the ^{19}F probe. We have only conducted the relaxation enhancement experiment for xamoterol as for other agonists the larger differences between the initial P5 and P2 intensities make a side-by-side comparison more difficult. However, based on the PRE results it seems a reasonable assumption to make that the similar shift changes of P5 observed for the ternary complexes in the presence of the other agonists or the apo receptor can be explained by the same environment effect. Accordingly, the outcome of the paramagnetic study is in further support of our initial suggestion where we postulated that the observed chemical shift changes upon receptor activation are indicative of the ^{19}F probe changing from a hydrophobic environment into a more solvent-accessible environment. This further

strengthens our interpretation in the context of changes in a continuously developing water pathway that allows TM7 to become more solvent accessible as shown in Fig. 5. Although we did not conduct a comparable experiment for the TM6 environment we propose that in view of the similar size of the shift changes TM6 might undergo a similar change in environment.

We have now included the outcomes of these new Gd³⁺ experiments into our revised manuscript in the Results section on pg 10, in the Discussion on pg 15 and the Methods on pg 24. We have added a new Supplementary Fig. 11, which shows the titration spectra and R_2 data including quantitative analysis.

3) Further, an alternative explanation that should be explored more is the possibility that the chemical shift difference observed for the tertiary complex is due to the proximity of side chains of the nanobody to the ¹⁹F NMR probes. Upon examination of a published structure of β 1AR in a complex with a nanobody and agonist (PDB 6H7N), this reviewer noticed that Ala 282 of β 1AR is only 3-4 Å away from the nearby residue Ser 27 of the nanobody. Also given that TET would further extend the side chain of the modified Cys at position 282, it seems reasonable to suggest that direct interactions between the nanobody and TET-labeled Cys 282 could account for the observed chemical shift changes of the new signal for tertiary complexes. Indeed, Phe 29 of the nanobody is also then potentially close enough that ring current effects from the nanobody may play some role. The same argument applies also to the ¹⁹F NMR probe in Helix VII, which also appears to be nearby several side chains of the nanobody. Also, a figure of the tertiary complex of β 1AR with agonist and nanobody showing the location of the NMR probes should be included in the text.

We thank the reviewer for this suggestion. We had initially considered the possibility of including this discussion on side chain proximity related effects already at the time of manuscript writing. Based on the PDB ID 6H7N structure coordinates for TM6 A282C^{TET} one could maybe have argued in favor of a side chain proximity effect from nearby residues located in the nanobody, but we realized that for TM7 such effects should be rather small due to the increased distances to ^{TET}C344. Further, as effects due to the proximity of non-aromatic side chain groups on ¹⁹F chemical shifts are little understood, we decided to drop this alternative explanation entirely from the original manuscript.

However, the suggestion from the reviewer has now made us reconsider this and we are grateful for the suggestion. In the revised manuscript we have now included a further section on the possible contribution due to proximity effects of the nanobody. To lead the discussion we have created an additional Supplementary Fig. 18 where we show the TM6 A282C (a) and TM7 C344 (b) environment, respectively, based on the coordinates PDB ID 6H7N of the ternary complex with xamoterol and Nb6B9. The distances from the Cysteine S to the closest residue side chains of the nanobody are indicated on the figure.

From Supplementary Fig. 18 (a) it can be seen that with 4.5 Å the relative proximity of S27 to A282C could contribute to a change in chemical shift while other residues seem to be too distant to be plausible contenders. For C344 S γ all nanobody residue side chains seem more than 8 Å away, making proximity related effects a rather improbable cause for the observed chemical shift changes.

However, as the S γ positions are unlikely to be representative of the real TET locations with their real orientations generally unknown, we extended our analysis considerably. We added a CF₃-CH₂-S- fragment to the S γ of A282C and C344 based on the structure PDB ID 6H7N to obtain a better estimate of the TET moieties and performed a 12 ns MD simulation with the receptor embedded in an equilibrated and hydrated lipid bilayer. The trajectories of the 12 ns simulations are shown in Supplementary Fig. 18c for A282C^{TET} and ^{TET}C344, displaying the distances between CF₃ and the closest nanobody side chains as a function of simulation time. The simulation trajectories show that for A282C^{TET} the shortest distances during the simulation change overall very little. Figure (d) shows a representative snapshot of the A282C^{TET} environment taken at 1.0 ns. The distances to the closest residues S27 and F29 are substantially increased, and are now beyond 10 Å, when compared to the distances to S γ in the structure 6H7N (a), making a proximity contribution due to nanobody residues much less prominent than in 6H7N.

For the ^{TET}C344 environment the simulation trajectory in (c) shows that the Cysteine side chain with the ¹⁹F TET tag stays mainly in a position that is more distant from the nanobody but for 8% of the simulation time has switched into a conformation that is closer to the nanobody. Representative snapshots of these two main orientations are shown in (e) for the orientation with the closer CF₃ distance to the nanobody (snapshot taken at 1.0 ns) and in (f) for the predominantly populated orientation (92%) with CF₃ more distant from the nanobody (snapshot taken at 10.4 ns). For both orientations the shortest distances are indicated. In the closer orientation (e) a distance of 4.6 Å to I104 is found, with all the other side chains too distant to be relevant. In contrast, in the more distant orientation (f) all distances are larger 8 Å. From this extended analysis using the TET moieties residue I104 would be the only one close enough to be considered for a proximity effect. However, the simulations suggest at the same time that this closer orientation is only adopted for 8% of the time, which must dramatically reduce a potential effect.

In conclusion, after assessing the distance situation using S γ positions based on the 6H7N structure as well as extending the TET side chains and performing a 12 ns MD trajectory of the receptor we did not find any conclusive evidence of residues that would substantially indicate that proximity effects would explain the observed changes in ¹⁹F chemical shifts upon ternary complex formation.

We are describing all of this now in the revised manuscript text on pg 15 & 16 together with the use of Supplementary Fig. 18. We have also added a paragraph on the MD simulations in the Methods on pg 25.

Further to the reviewer's request we have now also included an overview of the ternary receptor complex as an additional Supplementary Fig. 17, which shows the relative position of the full nanobody to the ¹⁹F probes attached to TM6 and TM7.

4) How did the authors assign the NMR signals? Specifically, how did the authors rule out that C124, C133, and C302 did not contribute to the observed NMR signals? While these are buried in the hydrophobic region, labeling of such buried cysteines has been observed in the literature before (see, for example, Susac et al. *Angew. Chem.* 2015). As a control, the authors should show ¹⁹F data where C282 is replaced with a different amino acid and the "background" signals shown. Also, ¹⁹F data should be shown with C344 replaced by the endogenous Ser.

We agree with the reviewer that the possible labeling of additional cysteine positions needs to be carefully considered when preparing ¹⁹F labeled samples, in particular if the labeling reaction is conducted in detergent micelles. In our case we mutated already C163L and C85V to eliminate their background contribution ($\Delta 2$ constructs) as we had established that these two positions were prone to being labeled at low but noticeable levels when the TET labeling reaction was conducted in detergent micelles. With the exception of the C344 (β_1 AR-m-Cys $\Delta 2$) and A282C positions (β_1 AR-m-TM6-Cys $\Delta 2$) the two constructs used in this study are identical (this includes the position of all the remaining cysteine residues). For these two constructs β_1 AR-m-Cys $\Delta 2$ and β_1 AR-m-TM6-Cys $\Delta 2$ in the apo state we recorded a ¹⁹F NMR spectrum as shown in Supplementary Fig. 4a. For each construct recorded, a single peak was observed so that the individual spectra allowed direct and unambiguous assignment of the two labeling positions as C344 (blue) and A282C (light green), respectively. The assignments are based on the reasonable assumption that the propensity of TET to label any of the other cysteines is unchanged by the presence of either C344 or A282C in the receptor. Accordingly, should other cysteines be labeled then we would observe the same additional peaks in both spectra. However, since both recorded spectra result in a single signal and both show at different positions, they can only relate to ^{TET}C344 (blue) and A282C^{TET} (light green), which completes the assignment procedure. We would like to point out in this context that C344 is endogenous and not a Ser as the reviewer is mentioning.

Further, as the two signals of ^{TET}C344 and A282C^{TET} are non-overlapping any unwanted residual background labeling of other cysteines should be detectable through the pairwise comparison of the two spectra. However, as there are no additional signals we can conclude that the labeling reaction under the conditions chosen is highly selective and uniquely taking place at the positions C344 and A282C. In the Supplementary

Information we have now extended the description of this assignment procedure in more detail in the legend to Supplementary Fig. 4a. We have also included a sentence describing this in the main text on pg 6.

5) How was the functionality of the proteins used in the current study determined? The authors state “The functionality of the construct was shown previously”, referencing previous work (Solt et al 2017), but in the Solt paper there does not appear to be new functional data reported. Rather, in the 2017 paper reference is made to the work by Isogai et al. that tested the functionality of a similar, but not identical, protein. Further, there does not appear to be any functional data presented in the current manuscript that demonstrate the modified constructs used for the current study bind ligand and are functional. Some data should be included to show that the proteins used for 19F NMR are functional (e.g. radioligand binding experiments of the proteins employed in the current study).

We thank the reviewer for this question.

i) We have not been able to successfully establish a reliable radioligand binding assay in our laboratory with our receptor constructs solubilized in LMNG. Accordingly, our collaborators (Chris Tate lab, LMB MRC Cambridge), which are leading experts on β_1 AR and other GPCRs, helped us on several occasions and conducted ^3H DHA (dihydroalprenolol) binding assays using sephadex G25F spin columns with 0.0001% LMNG. However, following optimization of dilution factors only very small amounts of functional receptor were detected (2%). It was suspected that there was a problem with running G25F columns with LMNG. Diluting the LMNG solubilised receptor into DDM and using G25F columns equilibrated with 0.1% DDM gave a slight improvement to just under 10% of the theoretical maximum but never resulted in showing sufficiently convincing amounts of functional receptor either. Dr Tate and team say that even under the best conditions with the most stable receptors they never manage to get more than 50% of functional receptor in radioligand binding assays and this is despite the fact that these receptors couple in much larger quantities to mini-G_s protein (i.e. that much more functional receptor is present in the samples than revealed by the radioligand binding assay). In the context of the reviewer query we have recently set out for another series of attempts but again have failed to obtain values that are closer to the theoretical maximum. The assay does not seem to work and we are not sure why, but this is not related to a compromised state of the receptor. The assay fails to inform on the amount of available functional receptor and very dramatically underestimates the amount of functional receptor present. With this we abandoned our trials of implementing a radioligand binding assay and concentrated on alternative means.

Although we have struggled on multiple occasions to set up a working radioligand binding assay the functionality of the receptor constructs used in our studies have been established by a range of other independent methods (i) to (v):

i) We can show the functionality of our receptor constructs through independent ternary complex formation by binding of nanobody. SEC analysis shows proof of ternary complex formation through a shift to a smaller elution volume when compared to apo or ligand-bound receptor. This can be further supported by SDS-PAGE analysis of the SEC peaks, which indicate both the presence of receptor and nanobody in the shifted peak. SEC traces and SDS-PAGE analysis of the peaks are shown below and we have now included this evidence as Supplementary Fig. 21 in the revised manuscript.

β_1 AR ternary complex formation investigated by SEC and SDS-PAGE. 14 μ M β_1 AR-m-Cys Δ 2 solubilized in LMNG were incubated with 1 mM isoprenaline and 21 μ M Nb6B9. **(a)** SEC (Superdex S200 10/300) traces of isoprenaline-bound β_1 AR-m-Cys Δ 2 in the absence (blue), and in the presence of nanobody Nb6B9 (red), showing formation of the ternary receptor complex (peak 2). **(b)** SDS-PAGE analysis of the SEC fractions. SEC fraction peak numbers and SDS-page lane numbers correspond to each other. M, molecular weight marker; lane 1, β_1 AR; lane 2, ternary complex of isoprenaline-bound β_1 AR-m-Cys Δ 2 complexed with Nb6B9 nanobody; lane 3, unbound nanobody.

ii) Our receptor constructs all bind to an alprenolol affinity column. This affinity column is typically used by us prior to the preparation of the NMR samples to separate functional from non-functional receptor and is highly specific for functional receptor. This method is widely accepted in the β_1 AR community as a proof of receptor functionality through the ability of the receptor to bind ligand in the orthosteric binding pocket. Binding is highly specific and elution from the column is selectively taking place through addition of a second ligand (atenolol) that is added in excess.

iii) As mentioned above, all the constructs used in this study bind Nb6B9. Nanobodies recognize specific epitopes, which means that for Nb6B9 to be able to bind to β_1 AR, the receptor needs to have the correctly folded structure. This proves that our constructs are functional.

iv) As shown in our 2017 study (Solt et al 2017) the β_1 AR-Met Δ 5 constructs show characteristic ^1H , ^{13}C chemical shift changes upon ligand addition that are easily rationalised in the context of receptor functionality as they correlate with the efficacy of the bound ligands. By itself this is in principle a self-consistent functionality assay. The constructs used in our current ^{19}F NMR study differ from β_1 AR-Met Δ 5 by two additional mutations C163L and C85V (Cys Δ 2). We have conducted a comparable ^{13}C HMQC NMR study using the new constructs and they behave identically to the β_1 AR-Met Δ 5 construct used in the 2017 study. The spectra are shown below for apo and xamoterol-bound receptor states as representatives of a more comprehensive set.

Receptor functionality investigated by ^{13}C methyl methionine NMR. Comparison of ^1H , ^{13}C HMQC spectra of ^{13}C -Met $\beta_1\text{AR-Met}\Delta 5$ (used in the study by Solt et al 2017 (ref ¹⁷ in main text) and ^{13}C -Met $\beta_1\text{AR-m-Cys}\Delta 2$ with $^{\text{TET}}\text{C344}^{7,54}$ used in the current study, in the apo form and bound to xamoterol. Both receptor constructs show equal spectral appearance in the apo form and the same behaviour upon activation through binding of the partial agonist xamoterol. It is concluded that the two receptor constructs behave very similarly and that the presence of the Cys $\Delta 2$ mutation and TET tagging at C344 has no deleterious effect on the functional response of the receptor construct $\beta_1\text{AR-m-Cys}\Delta 2$. Experiments were recorded at 800 MHz (^1H) and 308 K. We have now included this data as Supplementary Fig. 22.

v) To further show evidence of the functionality of the constructs used we have also formed ternary complexes using mini- G_s as the cytoplasmic binding partner. Mini- G_s is a much more realistic analogue of the full G_s binding partner. We have investigated ternary complex formation for $\beta_1\text{AR-m-Cys}\Delta 2$ using ^{19}F NMR. The formation of the ternary mini- G_s coupled complex is clearly visible. The ^{19}F NMR spectra are shown below. The figure includes further spectra of the apo receptor, and the receptor coupled to Nb6B9 with isoprenaline bound. We are currently conducting these investigations into ternary complex formation using mini- G_s in our laboratory and this will be the focus of a future publication. Accordingly, we do not want to include this material in the current manuscript. We provide this information solely for the review process and we would like to ask for this material to be considered in confidence.

^{19}F NMR spectra of various states of $\beta_1\text{AR-m-Cys}\Delta 2$ with $^{\text{TET}}$ C344 recorded at 308 K. Ternary complex formation with mini- G_5 for the adrenalin-bound receptor is shown in purple, while corresponding ternary complex formation with Nb6B9, with the receptor bound to isoprenaline is shown in blue. Isoprenaline-bound (green) and apo state (red) receptors are also presented.

We believe that with this, next to our ^{19}F NMR focused study, through (i) to (v) we have shown sufficient independent evidence that corroborates the functionality of the constructs used in our current study.

6) The noted observation that the conformational equilibria of the helix VI probe and helix VII probe are at least semi-independent is interesting and analogous to similar observations made for receptor-ligand complexes in earlier ^{19}F NMR studies of $\beta_2\text{AR}$ (e.g. Liu et al Science 2012, Eddy et al Structure 2016). The text describing the current observations (middle of page 16 in the Discussion) would benefit from discussing the present observations in the context of this literature data.

We have now included a clearer statement on the analogous observations made in Liu et al 2012 (our Ref. 18) and Eddy et al 2016 (our Ref. 52) and extended the discussion of our results in the context of their findings. Accordingly, we inserted the following text on pg 18 of the revised manuscript:

" ^{19}F NMR studies of the $\beta_2\text{AR}$ have previously reported a semi-independent response of the TM6 and TM7 conformational equilibria following binding of orthosteric ligands of differing bias^{18,52}. It was postulated that arrestin biased ligands preferentially activated TM7 over TM6, suggesting involvement of TM7 in biased signaling. Our ^{19}F NMR data for the ligand-bound $\beta_1\text{AR}$ do not show such a response, possibly hindered through the inaccessibility of the P3 signal. In contrast, we observe a ligand dependent variability of the signal position for TM7 in ternary complexes of $\beta_1\text{AR}$ (Fig. 4b) that seems largely decoupled from the response of TM6. As for the $\beta_2\text{AR}$ this might indicate therefore that TM7 in $\beta_1\text{AR}$ plays a role in signal bias."

We believe that this succinct addition clarifies the positioning of our findings in the context of the results shown in the two studies by Liu et al (2012) [our Ref. 18] and Eddy et al (2016) [our Ref. 52].

Additionally, the following minor points and questions should be addressed in a revised version of the manuscript:

7) The title should be revised to more accurately reflect the study. Specifically, the use of “GPCR signaling complexes” should be revised, as these aren’t actually signaling complexes studied but rather receptor complexes with small proteins that mimic some of the interactions observed with receptor-G protein complexes. I suggest the authors remove “signaling” from the title. This distinction is important especially as other groups (and likely the present authors) are also studying such receptor G protein complexes.

We thank the reviewer for pointing this out and we fully agree with him/her. Accordingly, we have removed the contested word ‘signaling’ and changed the title of our manuscript to “Conformational plasticity of ligand-bound and ternary GPCR complexes studied by ¹⁹F NMR of the β₁-adrenergic receptor”.

8) Supplementary Fig 2 – please include a reference to the PNAS Khorana paper where this scheme is first reported.

The reference to Klein-Seetharaman et al. PNAS 96, 13744-13749 (1999) is already cited in the main text e.g. on pg 4, but we have now also included it in the context of Supplementary Information Fig. 2 as requested by the reviewer.

9) Supplementary Fig 3 – the samples of β₁AR actually appear to be quite heterogeneous. This reviewer is somewhat surprised that such samples provided clean NMR data, though it does not call into question the quality of the presented data. Some comments from the authors on this in the text would be appreciated.

We thank the reviewer for raising this issue. There is a misconception here, as the SEC traces shown are not related to receptor material used for NMR investigations. The purpose of Supplementary Fig. 3 is mainly intended at emphasizing that the BTFMA TM7 labeled receptor did not interact with Nb6B9. These traces are clear evidence of very heterogeneous samples and we are quite sure that based on such materials we would have obtained very unfavorable looking ¹⁹F NMR spectra. In contrast, for all of our NMR investigations samples were additionally purified by alprenolol affinity chromatography prior to commencing any NMR work. A typical SEC profile of a TET labeled TM7 TETC344 sample recorded following prior purification via alprenolol affinity chromatography is now shown in a newly prepared Supplementary Fig. 4b. The affinity column separates functional from non-functional receptor. The excellent homogeneity of the sample consisting of pure and functional receptor can be fully appreciated. The quality of the sample shown is representative of all the samples that were investigated by NMR as part of this study.

Accordingly, the original Supplementary Fig. 4 has now become Supplementary Fig. 4a. In the figure legend to (b) we are pointing out that the size exclusion Superdex S200 10/300 chromatography trace shown is of apo β₁AR-m-CysΔ2^{TET}C344^{7,54} and was recorded after initial purification by atenolol affinity chromatography. We are emphasizing also that the high level of receptor purity and homogeneity displayed in the chromatogram is representative of all the NMR samples used in our study.

10) For the samples where no ligand is added (apo) and a nanobody is added: how does the signal intensity of the tertiary complex compare with the known reported basal activity of the protein?

We thank the reviewer for raising this interesting question. The answer to this relates to Fig. 4 where the ternary complex formation is shown as monitored by TM6 and TM7, in particular Fig. 4c,d where the amount of ternary complex formed is shown to correlate with the efficacy of the different ligands that are bound to the receptor.

Earlier measurements conducted in the Tate group (MRC LMB, Cambridge) using CHO cell lines stably expressing turkey β₁AR established an isoprenaline stimulated increase in ³H-cAMP accumulation of 22.5 ± 1.3-fold over basal (Sato et al. Mol Pharmacol. 88, 1024-1034 (2015). Setting the isoprenaline readout to a maximal response of 100% this relates to a basal (or constitutive) activity of the β₁AR receptor of just below 5% i.e. substantially lower than e.g. for β₂AR.

Before further addressing the reviewer's comment we would like to point out that in the original Fig. 4c,d the data point shown in light green for TM6 and TM7 of the apo receptor bound to nanobody Nb6B9 should have appeared at an efficacy of 5% and not as displayed at 0%. This was a mistake on our behalf and we have now corrected the position of this data point in both figures. We have repeated the linear fits taking into account the new values for the apo receptor. This only resulted in very minor changes without even affecting the R^2 values. In Fig. 4c,d the updated linear fits are now shown as dashed lines in black. As in the original version of the figure the fits are based on the available experimental measurements without enforcing an intersection with the (0,0) origin. As such, the correlations for TM6 as well as for TM7 are slightly offset to the right and result in relative signal intensities for the Apo state at a 5% efficacy value of 0.18 (TM7) and 0.21 (TM6) compared to 1.0 for isoprenaline, i.e. approximately reaching 20% of the isoprenaline value. A basal value of 20% is considerably higher than the 5% basal activity determined in the cellular assay and might point towards a slight increase of the amount of ternary complex formed at low efficacies such as in the apo form and when bound to low efficacy ligands due to the inherently high affinity of the Nb6B9 nanobody for the receptor. Unfortunately due to the low basal efficacy of the receptor finding a true inverse agonist for β_1 AR (turkey as well as human form) has not been successful in the β_1 AR field and accordingly we were not able to conduct an NMR measurement in the presence of a nanobody emulating a situation corresponding to 0% efficacy. However, one could argue that based on our model in a 0% efficacy situation there should be no ternary complex formed. We repeated the linear fits for TM6 and TM7, enforcing this time that they go through the (0,0) origin. These corresponding alternative linear fits are now also shown in Fig. 4c,d as dashed grey lines. The latter fits resulted in relative signal areas at 5% efficacy of 0.052 for TM7 and 0.058 for TM6, corresponding to 5-6% of the isoprenaline value of 100%. This is in excellent agreement with the 5% basal activity value determined in cellular assays from the stimulated cAMP increase. Therefore even at the lower efficacy levels such as for the apo form of the receptor the amount of receptor coupling to the nanobody relates very well to the cellular signaling response associated with the coupling of the heterotrimeric G protein.

We have now included both linear fits in Fig. 4c,d and added a comment to this in the figure legend.

11) All the presented NMR spectra were recorded at 308K. Were data at lower temperatures measured and, if so, is there any evidence of multiple components in the signal envelopes?

Representative experiments for TM7 ^{TET}C344^{7.54} for the apo receptor and bound to xamoterol were recorded at 288 K, 298 K and 308 K. Over the temperature range investigated we did not find any evidence of multiple signal components to the P2 signal in the spectra. An overlay of the spectra of apo receptor and for β_1 AR bound to xamoterol recorded at the temperatures 308 K, 298 K and 288 K has now been included as Supplementary Information Fig. 20. An investigation with TM6 revealed a similar situation but was not systematically done.

12) Supplementary Figure 8 – what is the x axis shown here? This should be labeled.

We thank the reviewer for having spotted this and apologize for this omission. The x-axis in Supplementary Fig. 8 is displayed in 'Hz'. The displayed spectral width of each spectrum is 2,000 Hz. We have now added this information to the Figure legend.

13) In several spectra there is a reported "degradation product" (e.g. spectrum of Iso + Nb6B9 in Figure 2 panel a). What is this specifically?

We thank the reviewer for this remark. In the legend to Fig. 2 we described the appearance of a hydrolysis product. However, this is strictly speaking not the correct way to refer to it and we have revised this in the text and figure legends. As such the peak marked with Δ shows the gradual appearance of small amounts of

free TET following the slow cleavage of the S-S bond at 308 K. We mention this occurrence now in the legends to Fig. 2, Fig. 4, Supplementary Fig. 6 and Supplementary Fig. 8 and we are further pointing this out also in the main text on pg 6. The rate of S-S cleavage is very slow compared to our experiment times so that this process does not affect our investigations. As the free TET signal is much sharper than the receptor signals, the peak tends to appear more prominent, while in fact the amounts of free TET remain very small. We would like to point out that the occasional presence of this peak does not interfere with our qualitative or quantitative data interpretation. Any longer duration experiment series such as the saturation transfer experiments or CPMG experiments were recorded interleaved with reference experiments in order to monitor the changes in the sample. We explain this already in the Methods section of the manuscript.

14) Figure 4 – my opinion is that this would be more graphically pleasing to visualize if the order of the presented spectra were reversed.

We thank the reviewer for this suggestion, which we completely agree with. We have now reversed the order of the spectra in Fig. 4a,b starting with the spectra of the lowest efficacy complex at the top.

15) Page 18 Discussion, “The cytoplasmic side of TM6 is well ordered, determined by the IBP coupling partner, shows few dynamics and reveals . . .” “shows few dynamics” is imprecise and not proper language for this manuscript. Please revise or remove completely.

We are grateful to the reviewer for having spotted this editing error. On pg 20 we have changed this now into ‘...shows no evidence of s-to-ms dynamics.....’.

Reviewer #3

We would like to thank the reviewer for his/her very favourable comments and for the strong endorsement and appreciation of our work. Below follows our response to his/her comments (for ease of handling we numbered the individual topics).

1) *The authors should comment why they (had to) use(d) a 33 to 50 fold excess of ligand over receptor to produce the (very small) chemical shift differences in the 19F NMR spectra.*

We thank the reviewer for raising this point. There seems to be an underlying misconception here, which we would like to clarify. The excess concentrations of ligand used are not necessary to induce the chemical shift changes but to ensure that the receptor is fully ligand bound throughout the NMR measurements (i.e. saturated with ligand), in order to minimize effects related to the presence of a mixture of bound and free receptor and to reduce effects related to the agonist on-/off-rates of binding that could strongly complicate the data interpretation should lower ligand concentrations be used. Using such a large excess of ligand is the established norm for these types of investigations and in the general context of binding studies is well understood. Accordingly, the conditions used by us are in complete agreement also with all the other NMR studies of GPCRs published in the literature. There is no indication for multiple or lower affinity secondary binding sites on the receptor, as this would clearly show in the NMR spectra. Hence there is no need to be concerned about the use of large excesses of ligands. In our previous work (Solt et al. *Nat. Commun.* 8, 1795 (2017), in particular Supplementary Table 11) we calculated the ligand-bound receptor populations for different agonists based on published ligand affinity K_D values (Baker et al. *PLoS One* 5(11): e15487 (2010)), sample protein concentration and ligand concentration. For every ligand used the population of ligand-bound receptor was >99.9%. For example for carvedilol >99.999%, for cyanopindolol >99.999%, for xamoterol >99.972% and for isoprenaline >99.988% of the receptor was present in ligand-bound form. This

is exactly what is required for our investigations as it shows the receptor de facto fully bound to the ligand. As this has been established in the literature a long time ago and similar experimental conditions have been used by others multiple times prior to our study we do not feel that there is any particular need in our manuscript to elaborate further on these well-understood issues.

2) The effective removal of the exchange broadening for C344 in the CPMG experiments is very intriguing. Did the authors also confirm that no relaxation dispersion is measured for site A282C?

We are grateful to the reviewer for his/her interest into the dynamics investigations and for raising this question. We did not perform a CPMG experiment on A282C^{TET,6,27} as obtaining a good quality dispersion curve with small experimental errors relies on the availability of sufficient amounts of protein. This is experimentally quite involved as the sample concentration needs to be on the order of 80 μ M or more. The exchange phenomenon mapped by the CPMG measurement relates to interchange between two inactive states I_1 and I_2 , which to some extent is a sideline of our main investigation and plays a lesser role in the general receptor activation scheme. All things considered, we therefore found it difficult to justify the demanding experimental undertakings, which would put us in the position to obtain a high quality CPMG measurement on TM6. Mostly, however, we believe the reviewer's question is already answered with the data provided in the current version of the manuscript:

We are in full agreement with the reviewer and see the question regarding TM6 as fully justified. As it turns out, a priori there is no need for a CPMG experiment to answer the reviewer's question. By simply considering the transverse relaxation rate constants R_2 in Supplementary Table 2 we can immediately see that exchange broadening can also be found in the corresponding P1 signal of TM6; however, to a lesser extent compared to the P2 signal of TM7. The R_2 values for the P1 signal of TM6 are increasing from the apo receptor to the isoprenaline bound receptor from a value of 115 Hz to 140 Hz. Accordingly, with the receptor bound to a full agonist we are observing an additional exchange contribution to the R_2 of the P1 signal of 25 Hz. This indicates indeed that compared to the apo receptor the full agonist bound receptor is more dynamic on the μ s-to-ms timescale. In contrast, the corresponding values for ^{TET}C344^{7,54} on TM7 (Supplementary Table 2) show an increase in R_2 of the P2 signal due to exchange broadening of 60 Hz, again comparing apo receptor with isoprenaline bound β_1 AR. Hence the exchange process affecting TM7 clearly has a more substantial effect on the R_2 ($R_2 \sim 1/\text{linewidth}$) of ^{TET}C344^{7,54}. Assuming that the same exchange process is affecting TM7 and TM6 then the larger exchange broadening contribution to R_2 of TM7 is the result of the more substantial difference in chemical shift between I_1 and I_2 of TM7. This agrees with the fact that for TM7 the P2 signal ($I_1 \rightleftharpoons I_2$) is gradually shifting towards higher field as higher efficacy ligands are added, while such a correlation is not really established for P1 of TM6. Accordingly, the peak positions of I_1 and I_2 for TM6 are much closer together (small $\Delta\omega$). In fact, their chemical shift positions are almost degenerate with the result that the additional contribution to R_2 exchange broadening of the P1 signal of TM6 is much reduced when compared to P2 of TM7. Of course, there is a possibility that TM6 might be affected by a different exchange process that has different exchange kinetics that might naturally result in a smaller exchange contribution to R_2 . However, the fact remains that the very small chemical shift difference $\Delta\omega$ between I_1 and I_2 of A282C^{TET,6,27} on TM6 will inherently result in a smaller exchange broadening contribution to R_2 than for ^{TET}C344^{7,54}.

Consequently, we do not think that there is need for a CPMG experiment to further assess the presence of an exchange process affecting TM6. TM6 is affected by chemical exchange as we can see by simply considering the R_2 values of A282C^{TET} that reveal the presence of exchange broadening. Compared to TM7 the effect is scaled down due to the close similarity of the chemical shifts of I_1 and I_2 on TM6. We explain these points already clearly in the Discussion on pg 11 of our manuscript but have added a sentence on pg 11 and on pg 12 to clarify that TM6 is also affected by a small amount of μ s-to-ms dynamics and that this relates to a small change in chemical shift between the exchanging states.

We have also added an additional statement in the Results section on pg 6 intended to draw the attention of the reader to the fact that the slight increase in P1 linewidth ($\sim 1/R_2$) observed when the receptor is

bound to full agonists clearly indicates the presence of an exchange process involving TM6.

In conclusion, both TM7, as well as TM6 are affected by μ s-to-ms exchange, but the latter is less accessible to our NMR study due to the insufficient chemical shift change. We believe that we have now sufficiently clarified this point in the revised manuscript.

3) I am not so sure about the occurrence of the Peak P6 of C344 in the presence of the nanobody. If I am correct, this is the position of the apo or agonist bound state of the receptor. As the nanobody is added, this peak shifts significantly, why would there be two conformations for the apo/agonist bound receptor in addition to the new conformation that is observed in the presence of the nanobody? Regardless, it would be good to show the deconvolution/simulation of these spectra with 2 peaks. Nevertheless, it is hard for me to comprehend that in P6, the receptor is also nanobody-bound as the downfield shift of the signal is completely missing. This conclusion should be reconsidered.

We thank the reviewer for raising this issue. As this topic is rather involved we feel compelled to provide a more comprehensive reply. In view of the relative complexity of the experimental evidence, we would like to commence by recapitulating our main spectroscopic observations of TM7 and their interpretation in the context of nanobody binding, leading to ternary complex formation. For the apo and ligand-bound receptor we observe a signal P2 (Fig. 2b, Supplementary Fig. 6b). The linewidth of that signal becomes broader due to increased exchange broadening when bound to higher efficacy agonists (Fig. 2d, Supplementary Table 2) and concomitantly this is accompanied by a small upfield shift of P2 (Fig. 2b-c, Supplementary Fig. 6b, Supplementary Table 2). Upon binding of the nanobody we observe the additional appearance of a new peak P5 noticeably downfield shifted by ca. 0.8 ppm (Fig. 2b, Fig. 4b), which we attribute to the formation of the ternary complex of the receptor. We assume that in P5 the cytoplasmic side of the GPCR with regard to TM6 is in its fully open conformation, allowing the docking of the nanobody. We name this state A^{G^+} and with regard to the position of TM6, this likely corresponds to the conformation observed in the crystal structures of class A GPCRs that is typically referred to as the fully active state of a receptor. In the manuscript on pg 14-15 we propose that the noticeable change in chemical shift from P2 to P5 is indicative of changes in water accessibility to the interior of the receptor in the ternary complex, to which our TM7 probe ^{19}F -C344^{7,54} is very sensitive. Our interpretation is underpinned by earlier MD simulations reported by Yuan et al. (*Nat. Commun.* 5, 1-10 (2014)), which showed a shift towards a more water accessible interior of the receptor in the active state. An interpretation of our observations is shown in our model representation as Fig. 5. In connection with a request from reviewer 2 we conducted additional PRE measurements using the water-soluble Magnevist Gd^{3+} compound. The results of a Gd^{3+} titration using TM7 ^{19}F -C344^{7,54} as probe showed that the P5 signal of the ternary form of the receptor is indeed more solvent-accessible compared to the P2 signal of the ligand-bound receptor form (Supplementary Fig. 11). This adds further support to our suggested explanation that changes in the hydrophobic environment in the vicinity of the TM7 probe as illustrated in Fig. 5 are a major contribution to the observed changes in chemical shift upon formation of the ternary complex.

Subsequently, we show further that the amount of P5 formed correlates with the efficacy of the ligand (Fig. 4b). In comparison to P2 the corresponding P5 signals are ca. 2.5 - 4 times broader. The R_2 values related to the linewidth are provided in Supplementary Table 2. Furthermore, a closer inspection of the 1D ^{19}F NMR spectra shows the presence of a third, additional broad signal that appears at a very similar position to P2. We named this signal P6 and due to the broad character of this signal and its overlap with P2 it is difficult to determine its exact peak position simply from the spectrum itself. Often, we refer therefore to the P2/P6 region. In fact, without the help of spectral deconvolution while measuring spectra only at one excess ratio of nanobody over receptor (here two-fold excess) it is easy to miss out on the presence of the P6 signal in the spectra. However, we deconvoluted all the ternary complex spectra as Lorentzian lines and as stated in the original manuscript it became clear that the signal envelope at the position of P2 under the conditions when forming the apo ternary complex in the presence of a two-fold excess of nanobody required two Lorentzian lines (Supplementary Fig. 8d); one with a linewidth corresponding to the apo receptor signal P2 ($R_2 = 75$ Hz) and one with a much broader linewidth ($R_2 = 300$ Hz) for P6. The linewidth of P6 is similar to

the one of the P5 signal ($R_2 = 320$ Hz) for the ternary state A^{G^+} . Accordingly based on the similarity in R_2 the initial suspicion arose at this stage that P6 might also relate to a ternary state, despite not showing the characteristic downfield shift seen for P5.

As requested by the reviewer in our revised version of the manuscript we have now repeated these deconvolutions to include all of the other ternary complex scenarios and find as shown in the revised Supplementary Fig. 8 that for carvedilol (e), cyanopindolol (f) and xamoterol (g) two Lorentzian signals are required for a satisfactory deconvolution of the spectral area centering on P2/P6. For every scenario the deconvolution resulted in a sharper component with an R_2 value similar to the corresponding P2 and a broad component P6 with an R_2 value similar to the one of the corresponding P5 peak. For isoprenaline we could not convince ourselves of an improved fit by deconvoluting with two Lorentzian lines. This was due to the limited signal-to-noise ratio in the spectrum considering the low intensity of the P2/P6 peak (Supplementary Fig. 8i). However, even when deconvoluting using a single Lorentzian for isoprenaline, it became apparent that the R_2 value of the fitting line had to be dramatically increased ($R_2 = 200$ Hz) beyond the one of the P2 signal ($R_2 = 140$ Hz), hence again indicating the presence of an additional broader P6 signal that overlaps with P2. In the revised manuscript we have updated Supplementary Table 2 with all the chemical shift and R_2 values for P2 and P6 obtained from the deconvolutions using two Lorentzian lines. These two-component deconvolutions of the P2/P6 region are shown in the revised Supplementary Fig. 8d-h, as mentioned above. In addition, we refer in the text on pg 9 and pg 19 to the new two-line deconvolutions that confirm the presence of the signal P6 that is overlapping with P2.

In the original submission we investigated further for the xamoterol case what happens if the excess of nanobody over receptor ratio is increased to five and ten times, respectively. As shown in Supplementary Fig. 9b it became evident in the spectra that the linewidth of the overlapped P2/P6 region was increasing as more nanobody was added. This underlined that overall more ternary complex P5 and P6 was formed, while the amount of the narrower P2 signal was decreasing. Accordingly, in the presence of more nanobody, the P2/P6 overlapped region became increasingly dominated by the properties of P6. For the case of xamoterol this was further confirmed by deconvolution with double Lorentzian signals, where an increase from a two-fold excess of nanobody (Supplementary Figure 8g) to ten-fold excess (Supplementary Figure 8h) showed an increasing amount of the broader signal.

Having firmly established at this point for the apo receptor as well as for all of the ligand-bound receptors that the appearance of this broad P6 signal was related to the presence of nanobody in solution, we reasoned further about the possible molecular assembly leading to this P6 signal. One plausible conclusion was that P6 might be related to the formation of a second ternary receptor complex. Initially we shared the reviewer's difficulty in comprehending how two ternary complexes might differ dramatically in their chemical shift positions such as P5 and P6 do. However, based on the similarity of the linewidth to P5, we postulated that P6 might also be a ternary complex, but where the nanobody is not bound to the active, open conformation of the receptor, thus not undergoing the substantial chemical shift change displayed for P5. Accordingly, P6 could be representative of a nanobody coupled ternary complex but with the receptor in a conformation that resembles an inactive form, explaining the similarity in the signal position to P2. In this complex the engagement of the loops of the nanobody binding the receptor might be incomplete and the interaction might be considered as pre-coupling resulting from the high affinity of binding of the nanobody. In our model in Fig. 5 we indicate this by showing the nanobody in A^{G^-} as fuzzy, emphasizing the incomplete engagement of the nanobody with the receptor. In line with these considerations of an inactive state we named this ternary state A^{G^-} . We are aware that a nanobody such as Nb6B9 is selective towards the active receptor conformation, however, this does not exclude the possibility of pre-coupling with the nanobody (and receptor) in an alternative orientation (conformation). Accordingly, the formation of A^{G^-} might not be representative of the situation for a G protein, where the affinity towards the receptor is much lower. We discuss this already in the manuscript on pg 19 & 20. In Fig. 5 we indicate further that the TM7 probe environment in the inactive ternary A^{G^-} is hydrophobic and likely very similar to the one experienced by the probe in $I_{1,2}$ thus resulting in similar chemical shift positions of P6 and P2. As further corroborated by the PRE experiments these chemical shift positions are relating to strongly reduced solvent-accessibility when compared to the more solvent-exposed and thus downfield shifted P5 signal of

the active ternary state.

As an alternative explanation for the observation of P6 we reasoned whether the nanobody binding event leading to the formation of A^{G^+} (P5 signal) might per se result in broadening of the P2 signal of the unbound receptor due to its exchange kinetics. As such P6 would in fact simply be an exchange broadened P2 signal. This would certainly agree with the high similarity between the P6 and P2 chemical shift positions. However, the established simultaneous presence of the sharp P2 and the broad P6 signals can't be reconciled with this scenario: In the case of an exchange process between unbound receptor and ternary A^{G^+} receptor the entire P2 signal would have to be broadened and appear as one single but broadened component. The latter would clearly contradict our experimental observations where a larger nanobody to receptor ratio led to increased broadening of the P2/P6 region (Supplementary Fig. 9b) and the results of the deconvolutions that clearly support the simultaneous presence of a sharp signal (P2) and a broad signal (P6). All of this is already clearly stated in various parts of the manuscript e.g. pg 9 and the Discussion on pg 19.

Accordingly, it is most likely that P6 is a second form of ternary complex that is inactive e.g. a pre-coupled state present due to the high affinity of the nanobody.

In the original manuscript we did proceed further to find out whether the signals P5 and P2/P6 were in direct exchange with each other. Using saturation transfer with the position of the saturating field at the center of P2/P6 we confirmed the signal area P2/P6 and P5 to be in exchange on a sub-second timescale (Supplementary Fig. 10a). To further discriminate whether the exchange involved P2 or P6 we conducted a second saturation transfer experiment where the saturating field position was moved away from P2 to minimize saturation, while at the same time still being able to saturate the broad tail of P6 (Supplementary Fig. 10b). Under these conditions the same exchange process was observed, showing identical exchange kinetics ($k_{ex} = 8.0 \pm 1.5 \text{ s}^{-1}$). This suggested that P6 is the signal in exchange with P5 and strengthens the suggestion of P6 as a ternary complex possibly representative of a pre-coupled receptor with conformational features similar to the inactive state, which we thus name A^{G^-} . The equilibrium between A^{G^-} and A^{G^+} is shown in our model in Fig. 5 where we have also indicated that in the earlier complex the nanobody is not fully engaged with the receptor, in contrast to A^{G^+} .

4) In principle I like the schematic hydrophobic/hydrophilic bar on Fig. 5. But I would not agree that in state A the probe sees a maximally hydrophobic and in state AG+ a maximally hydrophilic environment. In general, I do not understand why the authors do not picture the interior of the receptor as hydrophobic (gray), why is there a hydrophilic environment in the middle? Hydrophobicity and hydrophilicity are bulk properties, water channels and water mediated hydrogen bonds in my opinion cannot be considered as "phase".

We thank the reviewer for his/her comment on our final model Fig. 5. Next to providing an overview of the conformational interchange related to receptor activation, in Fig. 5 we aimed at reconciling our experimental observations with what we can say about the effects of the environment based on the chemical shift data of the probe on TM7. Our interpretation related to variations in solvent-accessibility is now further supported through the results of our recent Gd^{3+} PRE investigations that indicate increased water-access at the cytoplasmic region of TM7 in the ternary complex compared to the ligand bound state (Supplementary Fig. 11).

We realize that some of the shortcomings of Fig. 5 are related to the excessive use of blue on the inside of the receptor while showing a cross-section across the receptor TM6 and TM7 without sufficiently explaining the purpose of the blue color in the figure legend. Together this invited interpretations that were not intended by us and we are grateful to the reviewer for flagging up this shortcoming. In a revised version of Fig. 5 together with a more extensive figure legend we have aimed to improve the following points:

a) In Fig. 5 the bar below each of the states is not intended to provide an absolute measure of hydrophobic/hydrophilic character but to show a trend how the TM7 probe environments of the different states compare relative to each other. Accordingly, the ends of the sliders are not intended to suggest 0% and 100% and it can be considered that the sliders are open on both sides as we are not able to quantitate discrete values. The intention behind these sliders is solely to provide a relative comparison. We have now moved the position of the yellow spheres (representing our ^{19}F NMR probe) in the slider in the A state slightly away from the left and the one in the A^{G^+} state slightly away from the right. This way they should not be misunderstood as set to 0% or 100%. We hope that the reviewer finds this clearer now. We have also added a sentence in the legend to Fig. 5, which explains that this is a relative comparison across the different states. We prefer the descriptions of hydrophobic and hydrophilic environments according to the use of water-accessible/inaccessible but it infers the latter from its physical properties.

b) In our slightly revised color scheme of Fig. 5 the grey areas are still representative of regions that are dominated by their hydrophobic properties but we are now using two shades of grey to emphasize the differences between these regions. The light grey area is indicative of the lipid bilayer hydrophobic regions, while the dark grey areas, as shown prominently e.g. in I_1 and I_2 , reflect regions of the receptor that are rich in hydrophobic side chains and thus form a hydrophobic gate above and below the NPxxY region, shielding the inside of the receptor against the access of bulk water. The interpretation of the latter builds on the MD study by Yuan et al. (*Nat. Commun.* 5, 1-10 (2014)) as explained in the manuscript on pg 14. We have changed the color of these 'hydrophobic' gates to a different lighter grey to emphasize that these regions consisting of hydrophobic side chains (not lipids!) are different from the lipid environment of the bilayer. We explain this now in the figure legend.

c) Between the two hydrophobic gates we find internal water molecules. Previously this was shown as a blue region, which could lead to confusion and be interpreted as the presence of bulk water on the inside of the receptor. We have now modified this as shown in I_1 and I_2 to blue spheres representing individual water molecules as evidenced in high-resolution crystal structures of a range of receptors. The latter are ordered water molecules that are forming a hydrogen-bonded network and through the gates are kept separate from the bulk water. The blue circles are placed on a grey background to emphasize that the inside of the receptor is predominantly hydrophobic. The internal water molecules are kept separate from the bulk water through the gates and the latter prevents the formation of a continuous water pathway.

Upon activation, conformational changes lead to the gradual opening of a continuous internal water pathway in A^{G^+} with cytoplasmic influx of bulk water. Such changes in water accessibility are now further supported through our PRE data which show that TETC344^{7,54} on TM7 in the ligand-bound state of $\beta_1\text{AR}$ is less susceptible to line broadening than in the ternary state upon addition of the water-soluble paramagnetic relaxation agent gadopentetic dimeglumine (Magnevist). We interpret this as the A^{G^+} state being more solvent-accessible than the $I_{1,2}$ and A state. Accordingly in Fig. 5 we are showing a gradual influx of bulk water molecules upon formation of a continuous water pathway in the ternary complex. The latter is graphically represented by smearing out of the water molecules in the region between the helices and the removal of the dark grey hydrophobic gate regions. We maintain a grey background to stress that the inside of the receptor is still hydrophobic. However, there is now clear exchange between water molecules on the inside of the receptor and the bulk water phase indicated in A^{G^+} by blue smeared out 'delocalised' water, in representation of a water pathway across the receptor. We are now carefully explaining all of this in the figure legend.

We hope that with these changes our main message is now better conveyed and thank the reviewer again for his/her suggestion.

REVIEWERS' COMMENTS:

Reviewer #1 (Remarks to the Author):

All of my comments and concerns from the original submission have been fully and comprehensively addressed.

Jan K. Rainey
Dalhousie University, Halifax, Canada

Reviewer #2 (Remarks to the Author):

Frei et al. have submitted a revised and improved manuscript that largely addresses the questions and points of concern that were raised upon the initial submission. The revised manuscript has clarified several important points from the original submission and, where requested, provided additional data that strengthens the authors' initial interpretations. I therefore have no additional concerns and support publication of the manuscript.

Regarding the authors' response to the question about testing the functionality of the protein, this reviewer appreciates the technical difficulties of recording radioligand binding data for receptors solubilized in detergent micelles. Rather, the request was initially made assuming the authors would record ligand binding data for the proteins used in NMR studies in isolated membranes from the host expression organism, which is considerably less difficult and more widely seen in publications (even for NMR studies). Nevertheless, given the accumulated data presented, the authors make a reasonable argument that the modified proteins employed in the present study retain activity – ligand binding and complex formation with partner proteins – that is qualitatively similar to the native protein and likely "good enough."

Reviewer #3 (Remarks to the Author):

The authors have put together a very convincing and thorough revision of their manuscript reporting very interesting data on the structural dynamics of the b1-adrenergic receptor. All issues I raised in my review have been very well addressed. I am happy to recommend publication of this manuscript in Nature Communications.